# A method for predicting hydrogen and oxygen isotope distributions across a region's river network using reach-scale environmental attributes

Bruce D. Dudley[1], Jing Yang[1], Ude Shankar[1], and Scott L. Graham[2]

[1]National Institute of Water and Atmospheric Research, 10 Kyle Street, Riccarton, Christchurch 8011, New Zealand
[2]Manaaki Whenua - Landcare Research, 54 Gerald Street, Lincoln 7608, New Zealand

*Correspondence to*: Bruce D. Dudley (bruce.dudley@niwa.co.nz)

**Abstract.** Stable isotope ratios (isotope values) of surface water reflect hydrological pathways, mixing processes, and atmospheric exchange within catchments. Development of maps of surface water isotope values (isoscapes) is limited by methods to interpolate point measures across river networks. Catchment attributes that alter surface water isotope values affect downstream river reaches via flow, but some attributes such as man-made dams are no more likely to affect nearby unconnected catchments than distant ones. Hence, simple distance-based geospatial and statistical interpolation methods used to develop isoscapes for precipitation and terrestrial systems are less appropriate for river networks. We used a water balance-based method to map long-term average $\delta^2$H and $\delta^{18}$O for New Zealand rivers, incorporating correction using catchment environmental predictors. Inputs to the model are national rainfall precipitation isoscapes, a digital elevation layer, a national river water isotope monitoring dataset (3 years of monthly sampling at 58 sites) and river environmental databases covering around 600,000 reaches and over 400,000 kilometres of rivers. Much of the spatial variability in $\delta^2$H and $\delta^{18}$O of New Zealand river water was explained using the initial combination of precipitation isoscapes and a simple water balance model. $\delta^2$H and $\delta^{18}$O isoscapes produced by subsequently applying residuals from the water balance model as a correction factor across the river network using regression kriging showed improved fits to validation data, compared to correction using ordinary kriging. Predictors of high importance in the regression included upstream lake and wetland area, which was not strongly spatially autocorrelated nationally. Hence, additional hydrological process information such as evaporation effects can be incorporated into river isoscapes using regression kriging of residuals. The resulting isoscapes have potential applications in ecological, hydrological and provenance studies that consider differences between surface water isotope values and those of other components of the hydrological cycle (e.g. subsurface runoff or local precipitation).

# 1 Introduction

Stable hydrogen and oxygen isotope measurements in surface water provide powerful tools for hydrology, ecology and food science; uses include identifying runoff sources to rivers, determining migratory pathways of animals, and tracing the provenance of food (Kelly et al., 2005; Cable et al., 2011; Gao and Beamish, 1999). Typically, the utility of tracers in mixing and source partitioning studies relies on tracer concentration in potential sources being sufficiently distinct, relative to the degree of statistical noise in the system (Fry, 2006). Therefore, it is important to know spatially explicit distributions of hydrogen and oxygen stable isotope ratios (hereafter isotope values, following delta notation (Coplen, 2011)) for water bodies. Isotope values of river water are strongly dependent on those of precipitation (Kendall and Coplen, 2001), mixing of water sources, and isotopic fractionation resulting from different hydrologic processes occurring at and below the land surface within their catchments. Variations in the isotope values of precipitation reflect the temperature at which condensation occurs and prior condensation as air masses are transported over land, with the general effect that:

1. Precipitation becomes more depleted in heavy isotopes at higher latitudes, and higher elevations (Dansgaard, 1964).

2. Precipitation inland tends to be more isotopically depleted than that falling at the coast (Dansgaard, 1964; Winnick et al., 2014).

3. At mid- to high latitudes precipitation isotope values vary seasonally with changing temperatures, becoming more enriched in summer and depleted in winter (Craig, 1961).

The isotope composition of precipitation at a site is also affected by the source and trajectory of water vapour in air masses (Jouzel et al., 2013; Crawford et al., 2013; Mcdonnell, 1988), and sub-cloud processes including re-evaporation (Risi et al., 2008; Winnick et al., 2014). Understanding of the processes controlling precipitation isotope values has aided development of precipitation and surface water isotope maps (Bowen and Revenaugh, 2003; Bowen et al., 2011), with resulting hydrological and cross-disciplinary applications (Vander Zanden et al., 2016; Jasechko et al., 2013).

A range of hydrologic processes occur at and below the land surface that alter the isotope composition of surface waters relative to the isotope composition of precipitation. These include fractionation during atmospheric exchange processes at ground, water and vegetation surfaces (Evaristo et al., 2015; Ehleringer and Dawson, 1992; Gat, 1996), temporal and spatial patterns of sub-surface flow (Gonfiantini et al., 1998), and mixing between isotopically distinct waters (Klaus and Mcdonnell, 2013; Mcdonnell et al., 1991). A key limitation of current approaches to modelling stable isotope values of river water is representation of these processes across river networks (Bowen et al., 2011; Brennan et al., 2016). Geostatistical interpolation methods and linear regression used for spatial modelling of precipitation are less appropriate for surface waters because some landscape attributes that alter surface water isotope values, such as dams and lakes that increase atmospheric exchange (Gibson et al., 2016; Craig, 1961), are transferred to reaches downstream, but have no impact on nearby unconnected river networks. Hence, isotope values of surface waters are controlled by a mix of geospatial dependencies and those 'dendritic' dependencies unique to river networks (Brennan et al., 2016).

The water balance model approach of Bowen et al. (2011) provides a method of mapping hydrogen and oxygen isotope ratios across river networks that incorporates variation that can be modelled with interpolation approaches, and variation that follows the 'dendritic' patterns unique to river networks. This method takes advantage of spatial and seasonal predictability of precipitation isotope values, evaporative water loss and catchment discharge. Spatial patterns of residuals between predicted (modelled) and measured (annual averages of measured stream water isotope values) are then interpolated using ordinary kriging to improve models of river water isotopes. Bowen et al. (2011) note that residual adjustment using ordinary kriging

only indirectly considers many of the landscape attributes that may affect stream water isotope values. These attributes include those that affect temporal and spatial patterns of sub-surface flow, such as catchment slope (Salama et al., 1993), and those that affect atmospheric exchange processes, such as the presence of dams and natural lakes (Halder et al., 2015; Gat, 1996). When spatial variability within the catchment is low and sampling density is high, ordinary kriging methods may be appropriate, however, New Zealand is an example of a country with relatively large distances between long-term river monitoring sites but high variation in relief and climate over relatively short distances. Under these conditions, using regression based statistical methods with databases of explanatory environmental characteristics, rather than simple distance-based methods, to explain hydrological variation across the reaches of river networks provides superior prediction of spatial variability (Snelder and Biggs, 2002; Hicks et al., 2011). Regression kriging is spatial prediction technique often used in soil mapping that combines spatial similarity with non-location predictors (Hengl et al., 2007; Keskin and Grunwald, 2018). This technique would seem appropriate for correcting surface water isoscapes, for which divergence of isotope values away from those predicted using a water balance approach can be explained by catchment characteristics that are not strongly spatially autocorrelated. In this paper we follow the spatio-temporal water balance method of Bowen et al. (2011) to derive long-term average water isotope values from New Zealand rivers, using precipitation isoscape coefficients of Baisden et al. (2016), gridded meteorological data from NIWA's virtual climate station network (Tait et al., 2006; Tait, 2008), and a digital elevation map (DEM)-based river network (Snelder and Biggs, 2002). We calculate residuals between modelled surface water isotope values and values measured at 58 long-term river water sampling sites across New Zealand. We then extrapolate these residuals across New Zealand using a regression method that incorporates explanatory environmental variables from the River Environment Classification database (Snelder and Biggs, 2002), Freshwater Ecosystems of New Zealand (FENZ) geodatabase (Leathwick et al., 2010), as well as GIS lake data as predictors. We compare this 'regression kriging' residual estimation with ordinary kriging (Cressie, 1992; Bowen et al., 2011). We employ the method above to develop the first complete isoscape for the rivers and streams of New Zealand.

## 2 Methods

### 2.1 Water balance model for estimation of river water stable isotope values

Our initial method to calculate long-term isotope values of river water closely followed that given by Bowen et al. (2011) to calculate their 'spatiotemporally weighted water balance model' and are only briefly summarised here. For each 5 km × 5 km grid cell, we estimated monthly runoff (Q) as the larger of precipitation minus evaporation (P-E), or 0.01 × P; the latter estimate (0.01 × P) provides an estimate of dry-season and dry-region runoff (Bowen et al., 2011). We estimated the annual isotopic flux of runoff from monthly-weighted values for each cell following Eq. (1):

$$\delta Q_m = \sum_{i=1}^{12} [(P_i - E_i) \times \delta_i] \tag{1}$$

Where $\delta_i$ is the isotope value of precipitation within the grid cell at month $i$. The monthly-weighted isotopic flux ($\delta Q_m$) is calculated by summing across all monthly values (i). Routing of gridded runoff started with a 30m gridded DEM of NZ. We pit-filled the grid using Spatial Analyst in ArcGIS 10.6, then calculated flow direction next for each grid cell using the d8 method (O'callaghan and Mark, 1984), which considers flow to one of 8 adjacent cells. Values of Q and $\delta Q_m$ in each grid cell were then accumulated downstream using the flow accumulation tool in ArcGIS.

**2.2 Model input data**

Gridded precipitation isotope values for New Zealand were generated using the monthly models of Baisden et al. (2016). We
applied the regression coefficients of Baisden et al. (2016) to the geographical and climate predictor data given in their study, including elevation, latitude, pressure, solar radiation, air temperature, vapor pressure, wind speed, soil temperature at 100 mm depth, and an estimate of potential evapotranspiration (PET). Daily climate variables, PET and precipitation were retrieved from the NIWA virtual climate station network (VCSN; (Tait et al., 2006; Tait and Woods, 2007)), covering the period 1997-2018. Precipitation-weighted means were produced for climate drivers and coefficients were applied to yield monthly $\delta^2$H
and $\delta^{18}$O values for precipitation. While Baisden et al. (2016) also used VCSN interpolated data in their regressions, the VCSN is regularly updated as new physical climate stations are added to or removed from the national climate network and as the spline models are periodically improved (Tait et al., 2006). Hence, we checked the results of our procedure by performing regressions between our modelled, amount-weighted monthly precipitation isotope values and measured values from the dataset of Baisden et al. (2016), comprising monthly collections from 51 sites across New Zealand between 2007 and 2010.
Actual evapotranspiration (AET), as an input to the water balance model, was estimated according to the model of Porteous et al. (1994) assuming a soil water storage capacity of 150 mm and a critical value of 50% of this capacity, above which AET was assumed to be equal to PET and below which PET was scaled linearly to zero. Digital elevation input data were sampled from a 30m digital elevation model of New Zealand at the respective VCSN locations. Catchment environmental characteristics were taken from the River Environmental Classification (REC; (Snelder and Biggs, 2002)) and FENZ
geodatabase (Leathwick et al., 2010).

A summary of contributing data sources is shown in Figure 1; Table 1 gives a summary of water isotope datasets used in model calibration and validation.

**2.3 Calculation and kriging of model residuals**

Modelled river water isotope values were compared to annual average values from 58 sites from the National river water quality network (NRWQN; selected to represent catchments nationally (Yang et al., 2020)). Design of the NRWQN is described by Smith and Mcbride (1990), while descriptions of physical (catchment), flow and chemical conditions at monitoring sites can be found in Davies-Colley et al. (2011), Julian et al. (2017), and Yang et al. (2020). Measurements from this network have been used to develop and calibrate a range of hydrological and water quality models (e.g. Alexander et al.
(2002), Elliott et al. (2005)). The residual calculations used in this study compared predicted flow-weighted annual average river water isotope values to averages of measurements. Measurements were flow-weighted monthly values at each site from April 2017-March 2020. Data from April 2017-March 2019 is published in Yang et al. (2020). For the current analysis, an additional year of data (April 2019-March 2020) was collected from all NRWQN sites following the methods of (Yang et al., 2020). Briefly, samples were taken monthly from each of the 58 sites, and river flow rates were recorded at the time when
samples were collected. All samples were stored in 100 ml tubes in insulated ice bins, keeping samples at approximately 0°C and in darkness in the field, and then frozen and stored at approximately -20°C in the laboratory and later thawed for analyses. Isotope analyses were conducted using an isotope ratio infrared spectroscopy (IRIS) on a wavelength-scanned, cavity ring-down spectrometer (WS-CRDS) model L1102-i (Picarro, Sunnyvale, California, USA). Any variation in isotope values resulting from diurnal patterns was minimised by collection of samples at similar times of the day. River flow rates were
recorded at the time when samples were collected.

The locations of monitoring stations were adjusted in ArcGIS to coincide with the nearest grid cell of the stream network. Modelled isotopic compositions were extracted from each of the grid cells corresponding to a monitoring site, and the differences between modelled and observed calculated.

To improve reach-scale predictions of water stable isotope values beyond the predictions of the water balance model, we used

and compared two kriging methods, Ordinary Kriging and Regression Kriging. These two methods were used to extend calculated residuals between simulated and observed stable isotopes at our 58 validation sites across the New Zealand stream network. Ordinary Kriging, following the method of Bowen et al. (2011), only relies on spatial similarity (i.e. spatial autocorrelation) of the residuals. Regression kriging relies not only on spatial similarity, but also non-location predictors. In this case, the non-location predictors are catchment environmental variables available at sub-catchment scale across the New

Zealand from the REC and FENZ geodatabases (Snelder and Biggs, 2002; Leathwick et al., 2010).

In Ordinary Kriging, the prediction is a weighted mean of the observations:

$$y_i = \sum_{k=1}^{n} \beta_k(i) y_k^o \qquad (2)$$

where $y_i$ is the prediction at location i, $y_k^o$ is the $k^{th}$ observations (k = 1, …, n), and $\beta_k(i)$ is the kriging weight of $y_k^o$ to $y_i$ which is a function of locations of i and observation k and depends on the spatial autocorrelation structure of the variable

(Hengl et al., 2007).

In Regression Kriging, the prediction is the sum of a simple regression (normally linear regression as shown below) and the result of Ordinary Kriging:

$$y_i = \sum_{j=1}^{p} \alpha_j x_j + \sum_{k=1}^{n} \beta_k(i) e_k^o \qquad (3)$$

where $x_j$ is the $j^{th}$ environmental variable (j = 1, …, p) and $\alpha_j$ is the corresponding regression coefficient, and $e_k^o$ is the residual at kth observation after the regression.

Application of regression kriging includes two steps. The first step is to apply a simple multivariable regression aiming to select the best independent non-location variables. An initial list of over 30 potential independent variables for the regression model were selected from the REC and FENZ geodatabases based on our understanding of the hydrological processes which

potentially influence stream isotope components, including geological factors (e.g. slope, elevation, aspect, drainage density and lake coverage), climate factors (e.g., precipitation and evaporation), and land cover (Yang et al., 2020; Yang et al., 2019). Table A1 includes descriptions of these variables. Because many of these variables represent characteristics of the upstream catchment, they provide a representation of catchment-specific upstream processes (such as evaporation from wetlands, reservoirs, and lakes) that may affect river water isotope values. However, given the relatively small number of validation sites

available (58), correlation analysis and stepwise regression were applied to reduce number of independent variables. From the list of independent variables in Table A1, five were selected in the regression analysis based on BIC (Baysian Information Criteria), following the "one in ten rule" (e.g. Harrell Jr (2015)), i.e. one predictive variable can be included for every ten sites in the dataset. In the second step, spatial autocorrelation is considered together with the five selected variables following Equation 3 to give the prediction. Similarly, since non-linear regression requires an increased number of coefficients to be

estimated we also used linear regression in Equation 3 to avoid overfitting.

The performance of ordinary kriging and regression kriging in predicting spatial patterns of residual values was assessed across the 58 long-term river monitoring sites using k-fold cross validation (Stone, 1974). In this study, we applied LOOCV and 10-fold cross-validation to assess the performance of the two kriging methods.


**2.4 Data for final model validation**

Final model validation used water isotope data collated from previously published studies of rivers across New Zealand (Lachniet et al., 2021; Kerr et al., 2015; Stewart et al., 1983; Marttila et al., 2017). The report of Stewart et al. (1983)

includes $\delta^2H$ (but not $\delta^{18}O$) samples taken from nearly 200 sites throughout New Zealand between 1966 and 1981. These are

largely single samples for rivers but include repeated sampling over several years at some sites that highlighted storm-to-storm and seasonal variation. The work of Lachniet et al. (2021) is based on a single, high spatial resolution sampling campaign across the South Island of New Zealand in 2016. Kerr et al. (2015) reported single samples taken from small rivers across an east to west transect through the Southern Alps, in the South Island. Marttila et al. (2017) conducted over two years

of monthly sampling from 7 river sites in a small area of the South Island. The literature validation dataset is provided as a supplementary file (File S1) and detailed in Table 1. Because this validation dataset contains largely synoptic sampling data, to increase the number of annual average values in the validation dataset we included unpublished $\delta^2H$ and $\delta^{18}O$ data from 9 river sites for which regional government staff have conducted monthly sampling over date ranges between 2017 and 2020. Water samples from these sites were stored and analysed following the methods of Yang et al. (2020) and annual average

values were used in regression analyses. These sites are listed as *Dudley et al. unpublished data* in File S1. Because the dataset of Stewart et al. (1983) contained $\delta^2H$ but not $\delta^{18}O$ data, the $\delta^2H$ literature validation dataset is considerably larger (418 sites) than that for $\delta^{18}O$ (231 sites), and the $\delta^2H$ literature validation dataset also contains a higher proportion of sites in the North Island, because the large dataset of Lachniet et al. (2021) is restricted to South Island rivers.

**3 Results and Discussion**

**3.1 Precipitation isotope values**

Our modelled estimates of long-term precipitation isotope values are a close match to those published in Baisden et al. (2016), from which we took regression coefficients; $\delta^2H$ and $\delta^{18}O$ values become increasingly enriched in northern and low elevation

areas of New Zealand, with the sharpest spatial gradients at the western edges of New Zealand's central mountain ranges, which face New Zealand's prevailing West-Southwesterly winds and dominant western moisture source. Regressions between modelled, amount-weighted monthly $\delta^2H$ values and $\delta^2H$ values in monthly precipitation collections (i.e. the 2007-2010 monthly collection time series of Baisden et al.) resulted in $R^2 = 0.45$ and RMSE = 10.1 ‰ for monthly predictions and $R^2 = 0.79$ and RMSE = 5.1‰ for amount-weighted average values for the full 2007-2010 timespan of field sampling. Regressions

between modelled, volume-weighted monthly $\delta^{18}O$ values and measured monthly precipitation $\delta^{18}O$ values resulted in $R^2 = 0.48$ and RMSE = 1.2 ‰ for monthly predictions and $R^2 = 0.80$ and RMSE = 0.55 ‰ for 2007-2010 average predictions. These fit statistics differ very slightly from those published by Baisden et al. (2016), showing improved monthly fits, but a poorer longer-term fit for $\delta^{18}O$. This is likely due to updates to the climate network data used for VCSN model input, and updates to the interpolation scheme as described above.


**3.2 River water stable isotopes modelled using the water balance method**

Modelled, uncorrected estimates of long-term river water isotope values for each of the 58 NRWQN stations ranged from -68.66‰ to -32.54‰ (national average -49.83‰) for $\delta^2H$ and -9.56‰ to -5.67‰ (national average -7.56‰) for $\delta^{18}O$. Average (2017-2020) measured river water isotope values at these same NRWQN sites ranged from -77.01‰ to -19.75‰ for $\delta^2H$

(national average -46.38‰) and -10.63‰ to -2.85‰ (national average -7.25‰) for $\delta^{18}O$. The uncorrected model explained 72% of the observed variability in $\delta^2H$ values between sites, and 69% of the observed variability in $\delta^{18}O$ values between sites (Figure 2). Notably however, the uncorrected model showed a tendency to under-predict both $\delta^2H$ and $\delta^{18}O$ in warmer northern and low elevation sites (i.e. at the less-negative end of the range). Model under-prediction for these stations is likely due, at least in part, to evaporative enrichment of surface waters. Since these 58 NRWQN stations extend across a large latitude and

elevation gradient, rivers sampled at northern and low-elevation sites flow through catchments with substantially warmer mean annual temperatures that are likely to have led to greater evaporative enrichment. Many of the sites in northern New Zealand

also have large areas of lakes and wetlands in their upstream catchments (Yang et al., 2020) contributing to greater potential for evaporation.

Over-prediction of both $\delta^2$H and $\delta^{18}$O was apparent at some sites (Figure 2). This pattern has a precedent in areas of the continental USA with high topographic relief, where the spatially-weighted model of Bowen et al. (2011), which did not consider seasonal differences in P-E, appeared to under-predict the contribution of isotopically depleted runoff from high elevations to rivers. In that study, over-prediction was ascribed to the use of a model that did not consider seasonal differences in evapotranspiration; this issue was largely resolved using a temporally weighted model that considered these differences. In our study, this issue was most prominent in the South Island of New Zealand, which is characterised by high topographic relief, and high-elevation recharge sources dominate the flow of the majority of large rivers. However, in our study, we use a water balance model that considers seasonal differences in P-E but does not appear to have fully resolved this pattern. Potential reasons for this are, firstly, that the precipitation model we use over-estimates the isotope values of precipitation at high elevations or in leeward, rain-shadow areas; of the eight sites where predicted $\delta^{18}$O values exceed average measured $\delta^{18}$O values by > 1‰ (Figure 2), seven are in alpine-fed rivers on the leeward east of New Zealand. Rivers and streams leeward of the Southern Alps show isotopically depleted values characteristic of spillover of orographic precipitation from the windward west of the Southern Alps (Stewart et al., 1983; Kerr et al., 2015). While the linear regression approach employed in the precipitation model we used does not directly adjust for such orographic effects, it does to some extent capture areas of low $\delta^2$H and $\delta^{18}$O values in eastern New Zealand (Baisden et al., 2016).

A second potential driver of model over-prediction of river water isotope values is surface water / groundwater interactions. Rivers in lowland, leeward New Zealand tend to gain a significant portion of their flow from groundwater (Yang et al., 2019). Where this groundwater is dominantly derived from high-elevation recharge, gaining streams and springs tend to be depleted in stable water isotopes (Stewart et al., 2018). The (uncorrected) water balance model we employ does not consider groundwater pathways; this may result in error particularly in areas where gains from or losses to groundwater are considerable.

### 3.3 Kriging of isotope residuals

For regression kriging, simple multivariable regression applied to $\delta^2$H residual and $\delta^{18}$O residuals, individually, gave a final list of five independent environmental variables selected on the basis of t statistics in the two linear regressions (Table 2). These variables are site elevation and average catchment elevation (Site Elev and usCatElev), Average catchment slope (usAveSlope), upstream annual rainfall variability (usAnRainVar), and upstream lake and wetland area (usLWArea). The reduced regression analyses explained over 60% of the of the variation in the isotope residuals; $R^2$ values were 0.6 and 0.66 for the $\delta^2$H residual and $\delta^{18}$O residual regressions, respectively. The importance of each environmental variable (2nd column in Table 2), ranked by t statistic, shows upstream lake and wetland area, upstream annual rainfall variability and average catchment elevation are the 3 most sensitive variables in both $\delta^2$H and $\delta^{18}$O residual regressions, although the order of importance differs between the two analyses. A possible cause for the higher ranking of upstream lake and wetland area in the $\delta^{18}$O regression is the greater sensitivity of the $^{18}$O component of water to kinetic fractionation effects than the $^2$H component (Craig, 1961; Gat, 1996). Other variables in Table A1 sensitive to isotope residuals that were not included in regression included climatic variables (e.g. annual average potential evapotranspiration, annual average actual evapotranspiration, annual average precipitation, and upstream solar radiation in summer). Including these variables could improve $R^2$ values for regressions (e.g. by including annual average precipitation as a predictor, $R^2$ for $\delta^2$H and $\delta^{18}$O residuals can reach 0.76 and 0.80, respectively). However, this led to overfitting and caused the spatial isotope prediction in the following steps to be unreliable. Once the above 5 environmental predictor variables were chosen, regression kriging was applied to estimate spatial patterns of residuals across New Zealand. The performance of ordinary kriging, and regression kriging were then evaluated with LOOCV and 10-Fold CV. The results of this assessment for $\delta^{18}$O residuals are shown in Figure 3. Both LOOCV and 10-

Fold CV led to similar results: regression kriging simulations are scattered around the 1:1 line while those of ordinary kriging are mostly close to a horizontal line with values around 0. There is little predictive ability for ordinary kriging, as indicated by $R^2$ values below 0.1 for both LOOCV and 10-Fold CV. Regression kriging achieved satisfactory results, with $R^2$ over 0.75 for both LOOCV and 10-Fold CV. These results indicate regression kriging can be reasonably used to estimate spatial distributions of isotope residuals for rivers. It is worth noting that multiple regression contributed around 0.66 and kriging contributed around 0.1 of the total $R^2$. This emphasises the importance of the multiple regression step in this method, i.e. the first part of equation 3.

### 3.4. Spatial distributions of model residuals

Spatial interpolation of isotope residuals was applied to the entire national river network. Residuals interpolated using ordinary kriging (Figures 4A and 4C) show regional patterns; the water balance model tends to overpredict $\delta^2$H and $\delta^{18}$O values in rain-shadow areas in the east of both the North and South Islands and underpredict $\delta^2$H and $\delta^{18}$O values in the northern and western regions of both islands. Residuals shown in Figure 4 are calculated as the modelled minus average measured $\delta$ values for individual river reaches, so positive residuals occur where measurements were on average more depleted than the model estimate. Lower isotope values in precipitation and rivers leeward of large mountain ranges are driven by fractionation during orographic precipitation processes (Stern and Blisniuk, 2002; Poage and Chamberlain, 2001), and the patterns observed across the Southern Alps indicate that the dominant source of water to rivers in leeward areas is precipitation carried over the Alps by winds (Kerr et al., 2015). The overprediction of $\delta^2$H and $\delta^{18}$O values (of the water balance model) in rain-shadow areas indicates either greater recharge from high elevations to river flow within these regions, or local differences between actual and predicted (modelled) precipitation isotope values. We suggest that at least some of the patterns of residuals for the water balance model (Figure 4) can be accounted as a correction of the precipitation isotope model of Baisden et al. (2016) with regard to west-to-east depletion of precipitation isotope values associated with orographic rainfall across the Southern Alps. The basis of this conclusion is shown in Figure A1, in which we have plotted residuals between measured and modelled $\delta^2$H values for small rivers across an east to west transect through the Southern Alps (Kerr et al., 2015). The precipitation isotope model (i.e. Baisden et al., 2016) appears to underpredict $\delta^2$H values to the west of the main divide, and overpredict them to the east. The patterns seen in Figure 2 and Figure A1 indicate that these prediction errors were improved by the regression-kriging correction, though not fully resolved. Spatial patterns of the strongest predictors are shown in Figure A2; we believe that relationships between residuals and some predictors may reflect issues with representation of orographic effects by the precipitation model. All predictors in Table 2 except upstream lake and wetland area show a strong west to east gradient; for example, areas to the east of the Southern Alps where the water balance model overpredicts river water $\delta^2$H and $\delta^{18}$O values have lower average catchment slopes and higher upstream annual rainfall variability than are present to the west the alps where the water balance model underpredicts river water $\delta^2$H and $\delta^{18}$O values.

The significance of upstream lake and wetland area in the residuals regression can, however, be attributed with some confidence to evaporation from surface waters with long durations of exposure to the atmosphere. Wetland systems and lakes are relatively well-spread throughout New Zealand (Cromarty and Scott, 1996); glacier-carved lakes, and artificial lakes used for hydroelectric power are common through much of the Southern Alps region of the South Island, while volcanic crater lakes are present particularly in the central-eastern regions of the North Island (Figure 1G). Notably, the presence of validation sites on rivers draining major lakes creates areas of residuals that differ from regional trends. For example, within the central North Island region validation sites are sparsely spread, and roughly half of the validation sites sit on major rivers draining two caldera lakes, Lake Taupo (616 km$^2$) or Lake Tarawera (41 km$^2$). Using ordinary kriging, these sites create an area of negative residuals (e.g. Figure 4C) that are applied as a correction to all nearby river reaches. This effect suggests that the application of simple distance-weighted kriging of residuals may be inappropriate for our validation dataset as discussed in section 3.3, where sites are sparse relative to both land area, patterns of lake distribution, and topographic relief. Regression kriging

appeared to account to some extent for regional scale effects; regional patterns of overprediction in eastern rain shadow areas are represented, as are patterns of underprediction in northern and western New Zealand. However, regression kriging does not appear to create a regional pattern of negative residuals associated with lake evaporation in the central eastern North Island (Figure 4D).


### 3.5 Final model validation

To validate the results from the original water balance model (uncorrected), ordinary kriging (OK-corrected) and regression kriging (RK-corrected), we compared model results to collated literature values from previously published studies of rivers

across New Zealand (Lachniet et al., 2021; Kerr et al., 2015; Stewart et al., 1983; Marttila et al., 2017), and unpublished data from 9 sites as described in section 2.4. The validation analysis required assigning a reach identification number from the REC to each record reported in the citations. Using this approach, linear regression models between RK-corrected isotope predictions and all corresponding literature validation data explained 75.4 and 71.3%, respectively, of the $\delta^2$H and $\delta^{18}$O variance. We note that data from 6 sites sampled by Lachniet et al. (2021) for which we could not confidently assign a reach

identification number were not included in these regressions. The fit for $\delta^2$H is shown in Figure 5C. Root mean squared errors (RMSE) were 6.2‰ for $\delta^2$H on 416 degrees of freedom and 0.67‰ for $\delta^{18}$O on 228 degrees of freedom. Datasets incorporating repeat sampling of stable isotopes in New Zealand river water are scarce; only 23 of the 418 $\delta^2$H records in the literature validation dataset represent annual averages of monthly sampling. The river samples from the South Island rivers dataset of Lachniet et al. (2021) were collected within the month of November, within the austral spring snow-melt period;

South Island river water isotope values at this time are generally below the annual average (Yang et al., 2020; Marttila et al., 2017). Hence, we would expect our model predictions at sites from high latitudes with snowmelt influence to be generally more isotopically enriched than our literature validation dataset. This effect appears strongest in the uncorrected model results. When smaller rivers (Strahler order <3 where larger seasonal fluctuations in isotope values would be expected) were removed from the analysis, the fit of the final model improved, explaining 80.0% and 74.2%, respectively, of the $\delta^2$H and

$\delta^{18}$O variance across the literature validation dataset, with root mean squared errors (RMSE) of 5.7‰ for $\delta^2$H on 329 degrees of freedom and 0.62‰ for $\delta^{18}$O on 175 degrees of freedom. The most appropriate sites for validation of our model were those 23 sites for which long-term monthly sampling records are available, enabling us to compare predicted and measured annual average values. Average values derived from monthly measurements made over several years are likely to 'average out' much of the temporal variation in river water isotope values which results from temporal variation in precipitation

isotope values, evaporation, and contributions from different flow pathways. At these sites, the model explained 90.6 % of the $\delta^2$H variation across the dataset. Root mean squared error (RMSE) was 2.99‰ for $\delta^2$H on 21 degrees of freedom (Figure 5D).

While comparison of modelled annual average values to the full validation dataset containing predominantly single-sample values from river reaches (Figures 5A-5C) shows relatively poor model fits compared to those for long-term measurements

(Figure 5D), these comparisons serve to demonstrate improvement from the uncorrected and ordinary-kriging-corrected models to the final, regression-kriging corrected model.

### 3.6 Patterns and drivers of final model performance


**Patterns of precipitation isotopes, runoff and evaporation from surface water are the dominant drivers of δ²H and δ¹⁸O values in rivers**

Across New Zealand, spatial and temporal patterns of $\delta^2H$ and $\delta^{18}O$ in precipitation and runoff are dominant drivers of $\delta^2H$
and $\delta^{18}O$ in river water (Figure 6, and see Yang et al. (2020)). The (uncorrected) water balance model, which explicitly represents these factors, explained much of the variance present in our long-term river water dataset. For example, major patterns of depletion in precipitation isotopes with increasing latitude and elevation were carried through into predictions for rivers and passed downstream using the REC river network (Figures 6A and 6D). However, systematic bias was apparent; when compared to measurements from river monitoring sites, the spatiotemporal water balance model showed a tendency to
underpredict river water isotope values at the enriched end of the range of validation measurements and overpredict at the depleted end of this range (Figure 2).

**Regression kriging residual corrections improve models of δ²H and δ¹⁸O values in rivers**

Across the rivers of New Zealand, a water balance model corrected using regression kriging of residuals provided improved fits to measurements, compared to a model that used ordinary kriging interpolation of residuals between validation sites. Patterns of residuals across the validation site network showed distinct patterns which we attribute to orographic rainfall effects (Kerr et al., 2015; Purdie et al., 2010), and the position of lakes and wetlands in upstream catchments (Yang et al., 2020; Halder
et al., 2015). The effect of lakes and wetlands is an example of how landscape processes control $\delta^2H$ and $\delta^{18}O$ values in rivers; three factors combine to make regression kriging a particularly appropriate method to represent these processes. Firstly, lakes (including artificial lakes and dams) do not cluster predictably across New Zealand, and secondly, their effects are confined to downstream reaches. Thus, a modelling approach that considers the dendritic nature of river networks in this correction step is likely to better account for this process than one which corrects based only on Euclidean distance (Brennan et al., 2016).
Using a term for upstream lake and wetland areas in reach-scale regression-kriging incorporates enrichment effects of these water bodies and transfers this effect downstream while allowing for mixing. Thirdly, distances between validation sites in the NRWQN network are great, relative to the likely downstream extent of enrichment effects from lakes and wetlands. This raises the likelihood that corrections based only on distance weighting will extend outside of the extent to where their true effect will take place. Representation of evaporation effects in the final ordinary-kriging and regression-kriging corrected models is shown
using the example of the Lake Tarawera region in Figure 7. River water observed in the NRWQN validation dataset at the lake outflow is enriched in $\delta^2H$ and $\delta^{18}O$ relative to upstream precipitation and spatiotemporal model predictions, which do not account for evaporation. Using ordinary kriging, the correction factor based on the residual at the lake outflow site is applied to all nearby reaches and extends outside the catchment boundaries. Using regression kriging correction, the residual correction follows the dendritic nature of the river system and is passed only to downstream reaches.
Both the ordinary kriging and regression kriging approaches to residual correction resulted in partial correction of bias associated with orographic rainfall effects, however spatial extents of the corrections in leeward New Zealand differ between the two kriging methods (Figure 4). This reflects the data available to predict residuals for both methods; while the ordinary kriging method in limited by the distribution of (and lack of) validation sites across high-elevation areas of southern New Zealand, using regression kriging the correction could be applied at reach scale via factors that co-vary with orographic effects
(e.g. annual rainfall variance and catchment elevation, Table 2). Sharp changes in isotope values of small rivers (Kerr et al., 2015) and snowfall (Purdie et al., 2010) observed across the main divide of the Southern Alps have been associated with orographic effects. That the regression kriging correction method can apply residuals using geographical and climatological

gradients that co-vary with orographic weather patterns may partly explain improved fits of this model to river water measurements nationally.


**Regression kriging is vulnerable to overfitting and extrapolation**

As mentioned in section 3.3, there are more than five environmental factors in the river network datasets that are spatially correlated with patterns of isotope residuals. However, it is dangerous to include all sensitive factors in regression kriging as it may cause the problem of overfitting, which is very common in regression analysis and machine learning. Machine learning

methods often uses cross-validation techniques to solve the overfitting problem. In this study, there are only 58 sites which is insufficient for application of either of data-intensive machine learning methods or nonlinear regression, which increase the number of regression coefficients to fit. Instead, we focused on the five most sensitive environmental factors, following the "one in ten rule". Our results indicate this rule was appropriate for improving model performance and avoided overfitting; the addition of one more predictor in the regression model caused a wider and unreasonable range of isotope estimates in

subsequent national predictions. Extrapolation was another problem we encountered in preliminary regression analysis, as the training dataset did not cover the entire range of values for all catchment characteristics. To nullify this effect, we excluded reaches from the final model where the environmental characteristics of those reaches were outside of the range of the training dataset (e.g. Strahler order 1 streams with upstream coverage of lakes and wetlands greater than 30% of the total catchment area). Solutions to overfitting and extrapolation could include adding isotope sampling sites, and increasing the range of

environmental conditions across sampling sites (e.g. setting up river monitoring sites with upstream wetland areas approaching 1), or, for extrapolation, removing reaches from the final isoscape which have values for predictor variables substantially outside the range present across validation sites.

**Improved understanding of patterns of precipitation isotope values can improve understanding of hydrological**
**processes across landscapes**

Spatial patterns of residuals, and predictors of residuals from the spatiotemporal water balance modelling approach could potentially be applied as an investigative tool to describe flow pathways, or evaporation processes (Bowen et al., 2011). Regression kriging of these residuals allowed us to identify a suite of catchment environmental variables that were the best predictors of spatial patterns of residuals. However, as described above, patterns of residuals shown in our study are driven to

some extent by systematic bias in high-elevation precipitation isotope input data. Various authors have observed that the origins of air masses contributing to precipitation in New Zealand, and their transit across mountainous regions contribute strongly to spatial patterns of isotopes in precipitation (Mcdonnell, 1988; Stewart et al., 1983; Baisden et al., 2016; Kerr et al., 2015). These factors are difficult to capture with empirical models of precipitation isotopes of the kind used in this study (Bowen, 2010), however potential improvements to the precipitation isotope model could be made by incorporating high-frequency and

event-based sampling, by using machine learning prediction methods, or, more simply, by adjusting linear model predictions based on spatial patterns of residuals (Bowen and Revenaugh, 2003; Nelson et al., 2021). Improved regional precipitation isotope input data would raise the visibility of hydrological fluxes (such as high-elevation-derived groundwater contributions to rivers) in the regression kriging correction steps of our method. Similarly, improved regional precipitation isotope models are likely to improve the performance of process-based isotope hydrology models, such as the Isotope-enabled coupled

catchment–lake water balance model (Belachew et al., 2016), that are designed to quantify hydrological fluxes (e.g. between atmosphere, rivers, lakes, and groundwater) using water isotope data.

**4. Final conclusions and implications**

We have successfully produced isoscapes of $\delta^2H$ and $\delta^{18}O$ for the river network of New Zealand by employing a water balance-based model, enhanced by a regression kriging method to capture catchment characteristics not accounted for by the simple water balance. Although the final model performs well, future research is needed to improve model input data and identify the processes underlying spatial patterns of residuals identified by regression kriging. Further inclusion of these processes which contribute to fractionation and source mixing will reduce the need for correction methods, which are descriptive of current

patterns, but may not have predictive value with changing climate and hydrological modification.

Distinct differences in spatial patterns of $\delta^2H$ and $\delta^{18}O$ in river waters to those of precipitation highlight the value of river isoscapes in cross-disciplinary research. Differences between precipitation and river water isotope values were particularly evident at low elevations; New Zealand's high central mountain ranges create strong elevation gradients in precipitation isotopes, and lowland regions receiving more isotopically enriched rainfall are intersected by alpine-fed rivers bearing

isotopically depleted water from high-elevations. In addition to the hydrological implications described above, quantification of isotopic values for water sources across elevation gradients may be of particular benefit to those studies that attempt to attribute organic material (such as sediment-bound organic material transported in rivers) to particular elevation bands (Upadhayay et al., 2017; Feakins et al., 2016). The river water isoscapes shown in this study are also likely to be appropriate for studies utilising $\delta^{18}O$ values in human tissues to determine historical migration patterns (e.g. King et al. (2021)); local

drinking water provides a good proxy for the majority of total water intake in humans (Guelinckx et al., 2016; Ehleringer et al., 2008). Development of reach-specific predictions for water stable isotopes may also benefit ecological studies that examine movement of aquatic organisms (e.g. Brennan et al. (2015), Soto et al. (2013)); fine-scale study of animal movements across river networks can benefit from maps of river water isotope variation across similarly fine spatial scales.

**Appendices**

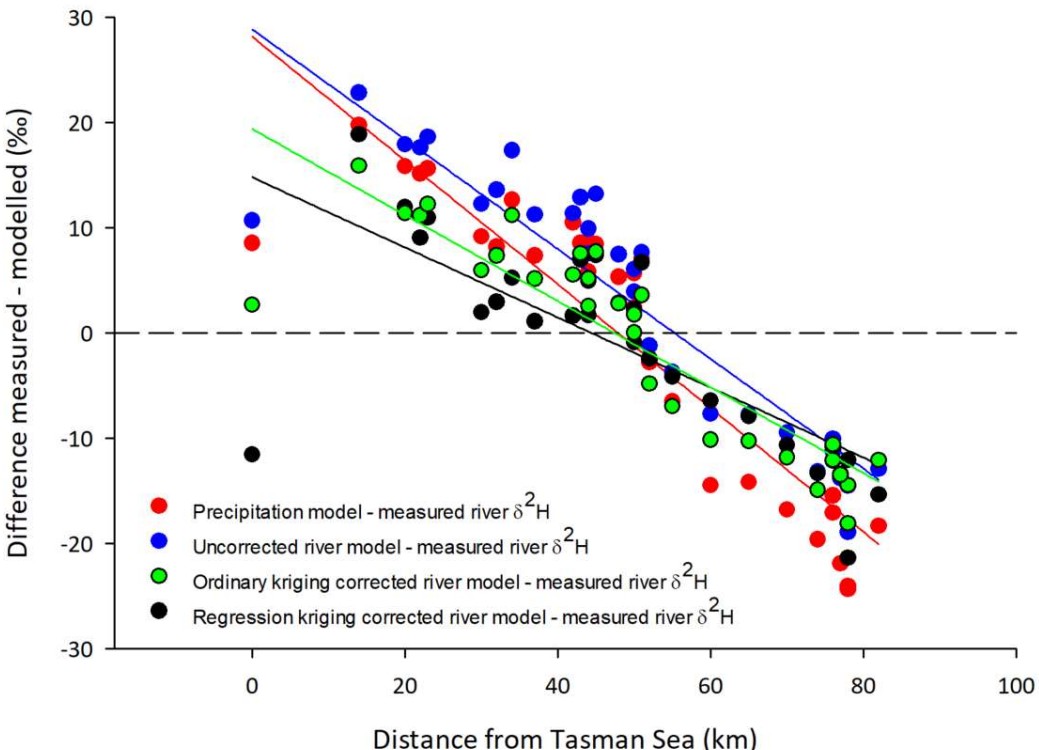

Figure A1. Comparison of measured and modelled water $\delta^2H$ values across a west-east transect through New Zealand's Southern Alps. Measured values are river samples, taken from small rivers by Kerr et al. (2015); modelled values are those for precipitation at the nearest virtual climate station network (VCSN) point, based on the coefficients of Baisden et al.

(2016), and uncorrected and residual-corrected river model values from this study. All model values show an

underprediction of $\delta^2$H values at the western (windward) end of the transect, and an overprediction at the eastern (leeward) extent of the transect. Root Mean Square Error for the four models across this transect are 13.8‰, 12.5‰, 9.8‰ and 9.6‰ for the precipitation model, uncorrected spatiotemporal river model, ordinary-kriging corrected river model, and regression kriging corrected river model, respectively.


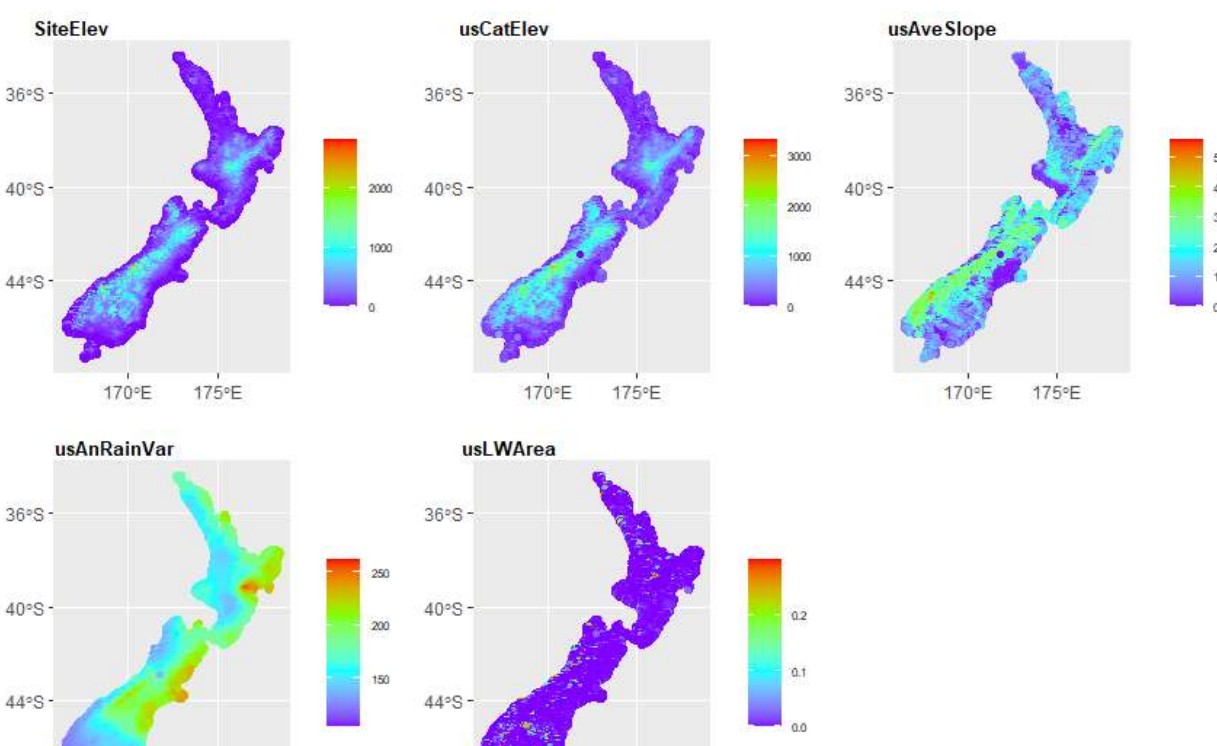

Figure A2. Spatial distribution of the five environmental variables used in regression kriging of residuals (Equation 3). Except "SiteElev", all other factors are averaged across upstream catchments.


Table A1. Full list of catchment environmental characteristic variables

| | Variable | Meaning | Source |
|---|---|---|---|
| 1 | SiteLat | The latitude of the reach (degree) | REC (Snelder and Biggs, 2002) |
| 2 | SiteElev | The elevation of the site (m) | Derived from 30m DEM (Snelder and Biggs, 2002) |
| 3 | CatMaxElev | Upstream maximum elevation (m) | FENZ (Leathwick et al., 2010) |
| 4 | CatArea | Upstream drainage area (km2) | FENZ (Leathwick et al., 2010) |
| 5 | PET | Annual average potential evapotranspiration (mm) | VCSN (Tait et al., 2006) |
| 6 | AET | Annual average actual evapotranspiration (mm) | VCSN (Tait et al., 2006) |
| 7 | Precip | Annual average precipitation (mm) | VCSN (Tait et al., 2006) |
| 8 | Dist2Sea | The distance to sea (m) : from the stream to the outlet stream | FENZ (Leathwick et al., 2010) |
| 9 | dsAveSlope | Downstream average slope (degree) | FENZ (Leathwick et al., 2010) |
| 10 | Order | A number describing the Strahler order of a stream in the stream network (-) | REC (Snelder and Biggs, 2002) |
| 11 | usAnRainVar | Coefficient of variation of annual upstream catchment rainfall (-) | FENZ (Leathwick et al., 2010) |
| 12 | usAveSlope | Average upstream catchment slope (degree) | FENZ (Leathwick et al., 2010) |
| 13 | usAvTCold | Average upstream temperature in cold seasons (degree) | FENZ (Leathwick et al., 2010) |
| 14 | usAvTWarm | Average upstream temperature in warm seasons (degree) | FENZ (Leathwick et al., 2010) |
| 15 | usFlow | Upstream annual flow (m3/s) | FENZ (Leathwick et al., 2010) |
| 16 | usLowFlow | Upstream mean low flow (m3/s) | FENZ (Leathwick et al., 2010) |
| 17 | usSolarRadSum | Upstream solar radiation in summer (W/m2) | FENZ (Leathwick et al., 2010) |
| 18 | usSolarRadWin | Upstream solar radiation in winter (W/m2) | FENZ (Leathwick et al., 2010) |
| 19 | usCatElev | Upstream catchment average elevation (m) | FENZ (Leathwick et al., 2010) |
| 20 | usLowGrad | Proportion of catchment with slope >30° (steep) | FENZ (Leathwick et al., 2010) |
| 21 | usLWArea | percentage of catchment in LCDB category (lakes, and inland and coastal wetlands) (%) | FENZ (Leathwick et al., 2010) |
| 22 | E_P | Ratio of evaporation over precipitation of the upstream catchment (-) | Calculated based on AET and precipitation from VCSN (Tait et al., 2006) |
| 23 | PET_P | Ratio of potential evaporation over precipitation of the upstream catchment (-) | Calculated based on AET and precipitation from VCSN (Tait et al., 2006) |
| 24 | DrainDsO1 | Drainage density for Strahler order 1 catchment | Derived from 30m DEM (Snelder and Biggs, 2002) |
| 25 | DrainDsO2 | Drainage density for Strahler order 2 catchment | Derived from 30m DEM (Snelder and Biggs, 2002) |

| 26 | DrainDsO3 | Drainage density for Strahler order 3 catchment | Derived from 30m DEM (Snelder and Biggs, 2002) |
|----|-----------|------------------------------------------------|------------------------------------------------|
| 27 | Dist2HeadO1 | Distance to Strahler order 1 headwater catchment (m) | Derived from 30m DEM (Snelder and Biggs, 2002) |
| 28 | Dist2HeadO2 | Distance to Strahler order 2 headwater catchment (m) | Derived from 30m DEM (Snelder and Biggs, 2002) |
| 29 | Dist2HeadO3 | Distance to Strahler order 3 headwater catchment (m) | Derived from 30m DEM (Snelder and Biggs, 2002) |
| 30 | Aspect | Average of upstream geographic aspects | Derived from 30m DEM (Snelder and Biggs, 2002) |
| 31 | Aspect_sd | Standard deviation of upstream geographic aspects | Derived from 30m DEM (Snelder and Biggs, 2002) |


**Code and data availability**

Modelled river water isotope values generated as described in this paper are available for download via NIWA's *NZ River Maps* site at https://shiny.niwa.co.nz/nzrivermaps/

A compiled literature validation data file used to produce Figure 5 is provided as supplementary file S1.

Precipitation and river isoscapes are available in GeoTIFF format as supplementary file S2.

Instructions for accessing and comparing datasets used in this work are provided in supplementary file S3.

River water measurements for model correction can be downloaded from the Global Network of Isotopes in Rivers (GNIR) database of the International Atomic Energy Agency (IAEA) through the WISER portal (Water Isotope System for Data

Analysis, Visualization and Electronic Retrieval).

Code used in this work is available on request from the authors.

**Author Contributions**

BDD, JY and US contributed to project conceptualization, methodology, investigation, data curation and formal analysis. BDD, JY and US contributed to writing and editing of the original draft. SG contributed to methodology, formal analysis, data

curation and editing of the original draft. BDD led project administration.

**Competing interests**

No competing interests

**Acknowledgements**

This work is funded by New Zealand's Ministry for Business, Innovation and Employment via the NIWA Strategic Science

Investment Fund (SSIF), and Te Whakaheke o Te Wai Endeavour Programme. We thank Russell Frew for access to data from the national precipitation isotope dataset reported in Baisden et al. (2016). We thank Troy Baisden and Liz Keller for advice on re-creation and implementation of their precipitation isotope model, but we note that any remaining errors are our own. We thank Amy Whitehead for uploading our results to NZRiverMaps and preparing the user guide in supplementary file S3. We thank all the many people who collected the water samples that we used to improve our model, and Kelsey Montgomery and

Oonagh Daly for sample analysis.

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

**Figures**

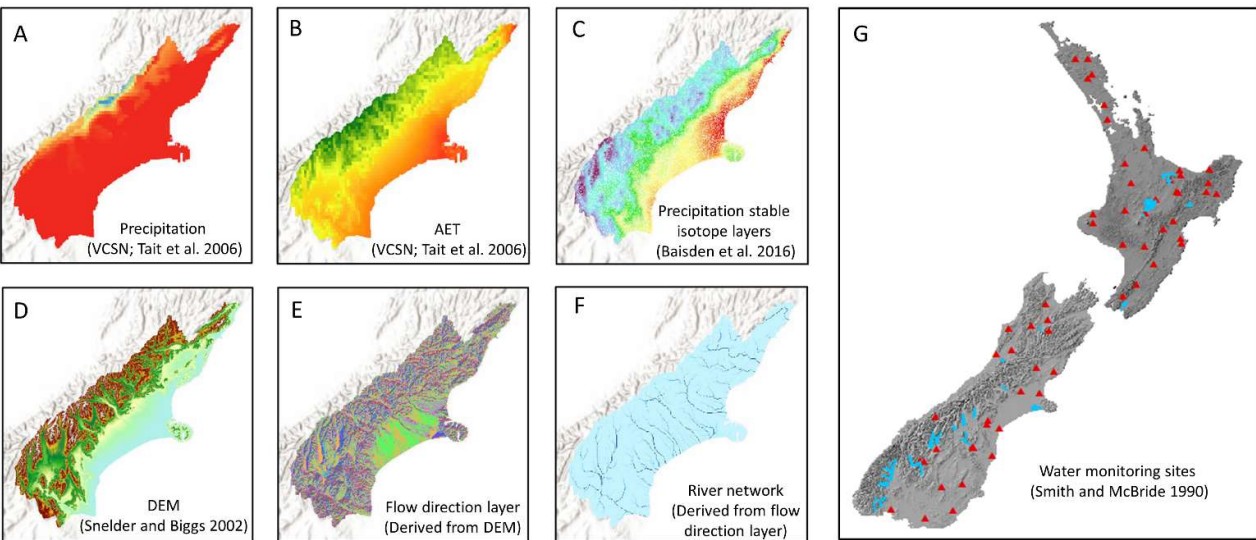

Figure 1. GIS layers and data sources used in model development. Panels A-F show GIS layers used for water balance modelling, plotted over the Canterbury region in New Zealand's South Island. Precipitation, Actual Evapotranspiration (AET), and stable isotope values of precipitation (panels A-C) are used to calculate catchment-scale flow-weighted isotope values of recharge ($\delta Q$, i.e. flow * isotope $\delta$ value)). The river network (F) is derived from the DEM (D) and flow direction (E) layers by setting a suitable flow accumulation threshold. Catchment $\delta Q$ values are routed downstream through the river network to

generate reach-scale estimates of $\delta^2 H$ and $\delta^{18} O$. Panel G shows river water isotope measurement sites (triangles) used for model correction, and major lakes (blue area).

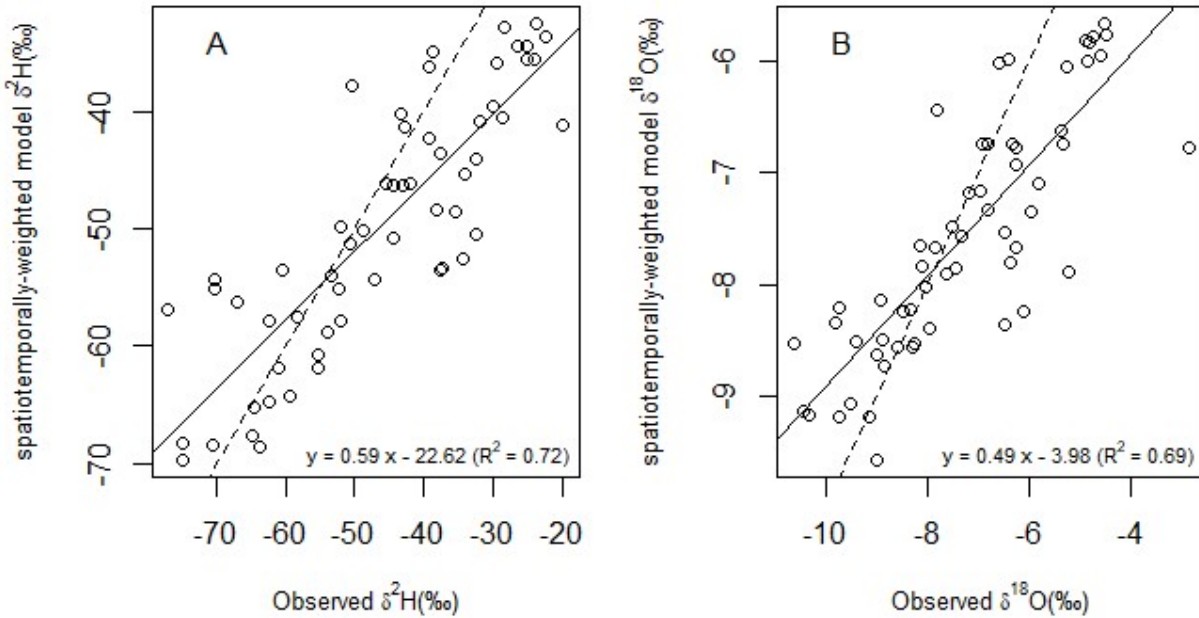

Figure 2. Relationship between modelled and observed δ²H (Panel A) and δ¹⁸O (Panel B) at 58 long-term river water monitoring sites across New Zealand. Modelled isotopes were the long-term averages of simulated isotopes computed following the water balance model approach of Bowen et al. (2011). The solid line shows the regression fit and the dashed line shows a 1:1 fit. The modelled tends to over-predict river water isotope at the more-negative end of the range (e.g. at high-latitude cooler southern sites) and under-predict at the less-negative end of the range (e.g. warmer, northern sites).


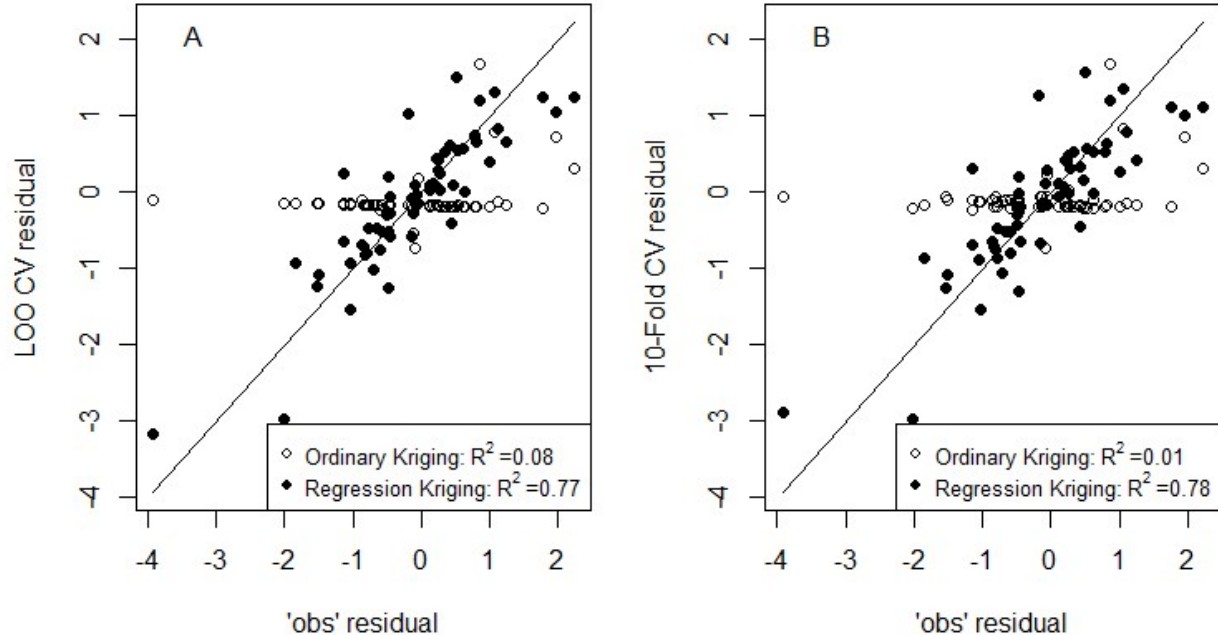

Figure 3. Comparison between observed residuals and spatial predictions of residuals made using two interpolation methods
– Ordinary Kriging and Regression Kriging. Observed residuals are differences between spatiotemporal model predictions of
$\delta^{18}O$, and measurements from 58 long term river monitoring sites across New Zealand. Comparison with estimates of residuals
generated using Leave-One-Out Cross-Validation (A) and 10-fold cross validation (B) suggests improved fit of regression
kriging, relative to ordinary kriging.

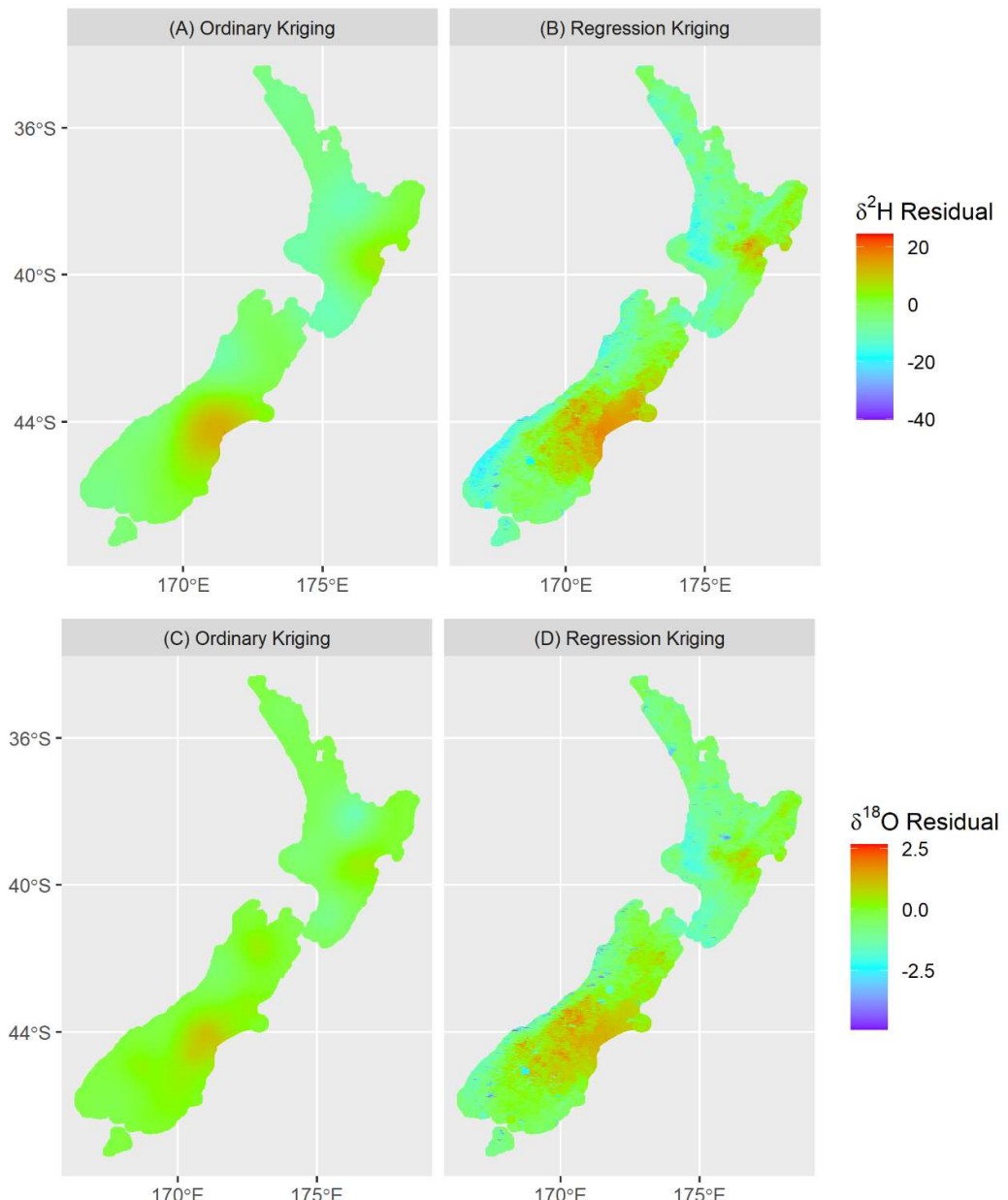


Figure 4. Distribution of isotope residuals (i.e. difference between modelled and observed isotopes) from ordinary Kriging (A and C), and regression Kriging (B and D). Positive values equate to a negative adjustment of river $\delta^2H$ and $\delta^{18}O$ values from the water balance model, as seen in leeward (eastern) areas of both the North and South Island. Ordinary kriging results in regional zones of negative isotope residuals around validation sites downstream from lakes and wetlands (such as in the central

North Island), while regression kriging predicts negative isotope residuals in low-elevation western areas, as well as in all lakes and wetlands and their downstream reaches.


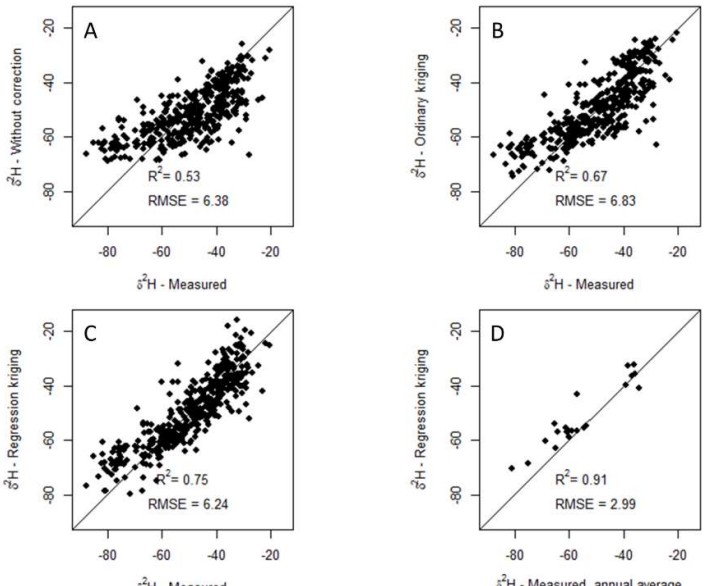

Figure 5. Comparison of literature values of water $\delta^2H$ across New Zealand rivers with predicted long-term average $\delta^2H$ values from: the uncorrected water balance model (A); ordinary kriging corrected model (B); and regression kriging corrected model (C). Panel D shows a comparison between predictions of the regression kriging corrected model with mean annual measured values at sites with > 1 year of monthly data. Lines show a 1:1 relationship.


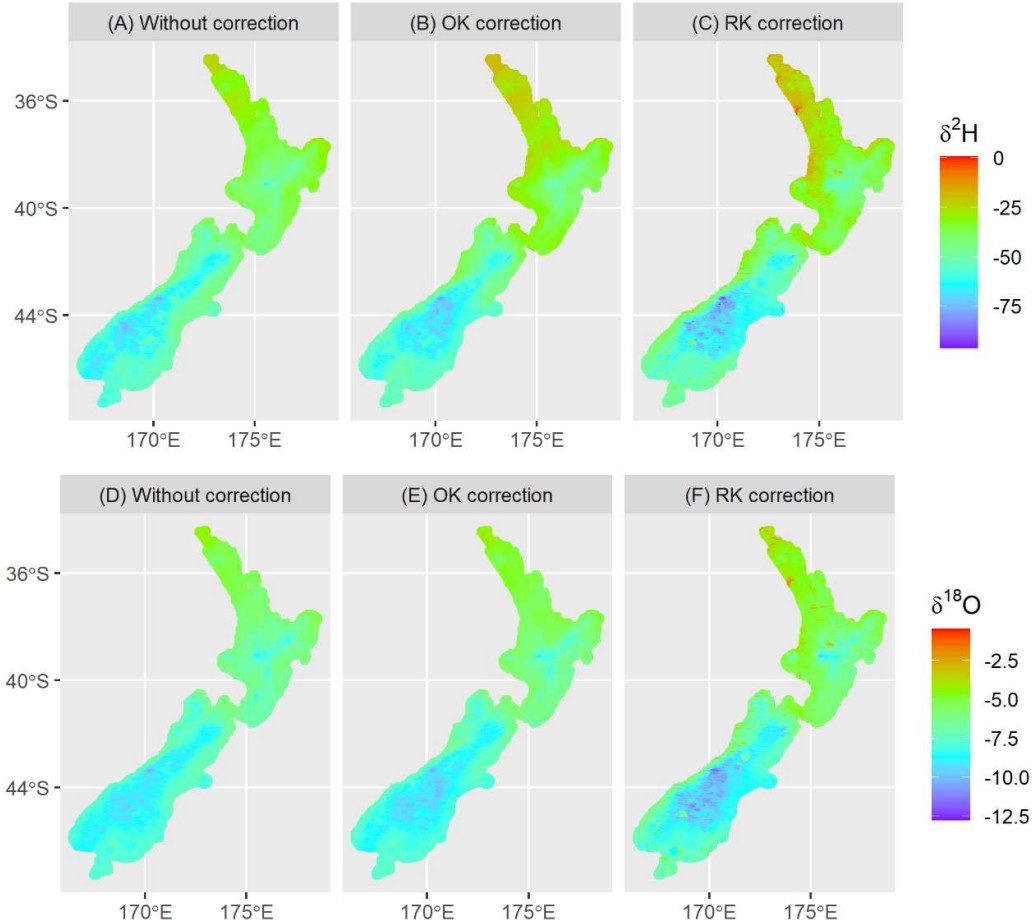

Figure 6. Predicted δ²H and δ¹⁸O of surface water of New Zealand. Panels A and D show uncorrected values from the water
balance model. Panels B and E gives residual-corrected values based on ordinary Kriging (OK correction). Panel C and F gives
residual-corrected values based on regression Kriging (RK correction).

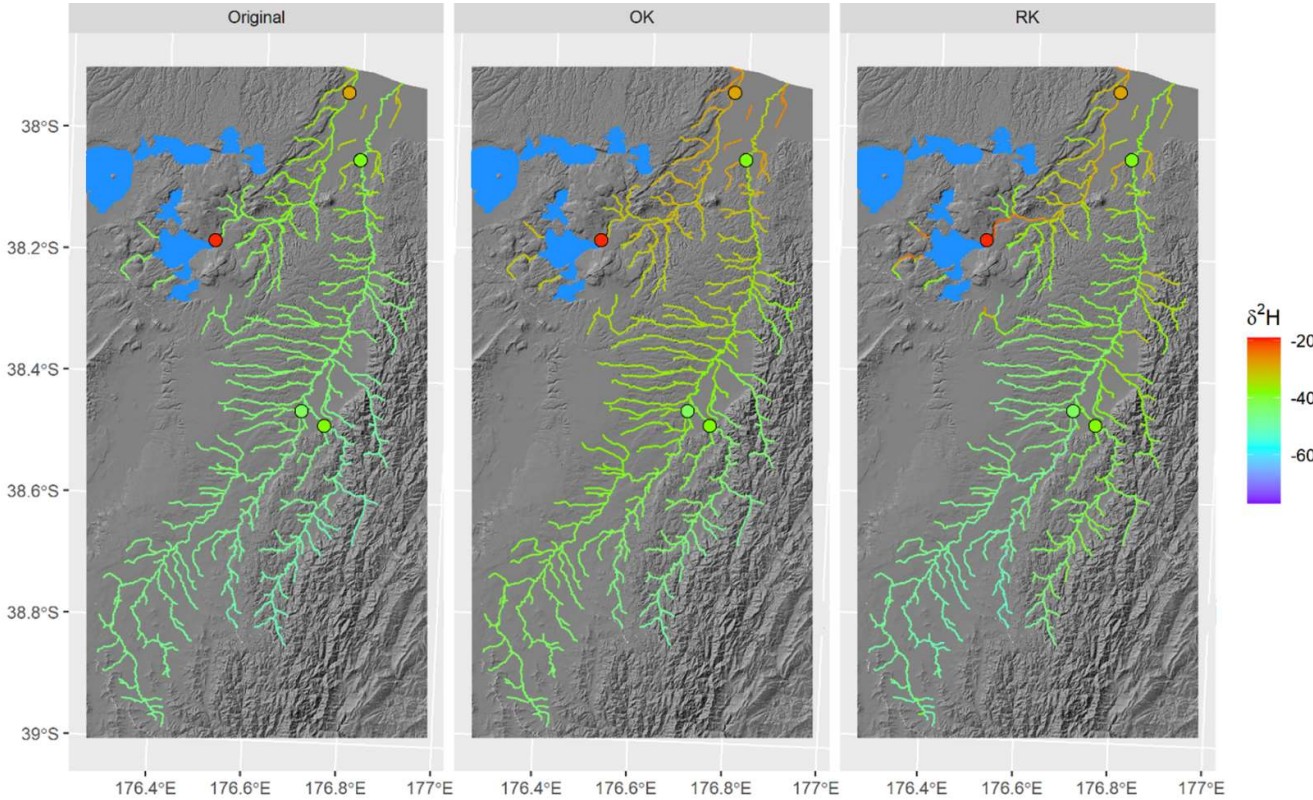

Figure 7. Comparison of modelled $\delta^2H$ values of river water across two neighbouring North Island basins (Tarawera basin

and Rangitaiki basin). Higher $\delta^2H$ values from $^2H$-enriched precipitation lower in these catchments are represented by the

water balance model (Original). Residual correction using ordinary kriging (OK) extends the enrichment effect of Lake

Tarawera (top left of each panel) across the entire region. Using regression kriging (RK), the lake enrichment is mostly

confined to downstream reaches. Points denote the locations and observed $\delta^2H$ values at the gauging sites.


**Tables**


Table 1. Water isotope datasets used in model calibration and validation. Lake samples provided in Stewart et al. (1983) were not included in our analyses. For Stewart et al. (1983) and Marttila et al. (2017), and unpublished data, annual averages for sites were used where available. Data used for river model validation is compiled in supplementary file S1.

| Model step | Data type | Frequency | Timespan | Number of sites | Sample count | Locations | Reference |
|---|---|---|---|---|---|---|---|
| Precipitation isotope model validation | Precipitation $\delta^2H$ and $\delta^{18}O$ | Monthly (accumulated sample) | 2007-2010 | 51 | 1457 | Throughout New Zealand | (Baisden et al., 2016) |
| River model calibration and cross validation | River water $\delta^2H$ and $\delta^{18}O$ | Monthly (grab sample) | 2017-2020 | 58 | 2051 | Throughout New Zealand (see Figure 1) | (Yang et al., 2020), this study |
| River model validation | River water $\delta^2H$ and $\delta^{18}O$ | One off (grab sample) | 2016 | 184 | 184 | Throughout South Island of New Zealand | (Lachniet et al., 2021) |
| | River water $\delta^2H$ | Various, depending on site | 1966-1981 | 188 | 750 | Throughout New Zealand | Stewart et al. (1983) |
| | River water $\delta^2H$ and $\delta^{18}O$ | One off (grab sample) | 2013 | 30 | 30 | South Island of New Zealand (West-East transect) | Kerr et al. (2015) |
| | River water $\delta^2H$ and $\delta^{18}O$ | Monthly (grab sample) | 2014-2016 | 7 | 176 | Waipara catchment, South Island | Marttila et al. (2017) |

River water δ²H and δ¹⁸O | Monthly (grab sample) | 2016-2018 | 9 | 166 | Southland and Manawatu districts, New Zealand | This study

Table 2. Selected environmental variables and related information, and importance ranks, p-values and t statistics in the regression for $\delta^2H$ and $\delta^{18}O$ residuals. Values for $\delta^{18}O$ residuals are in brackets.

| Variable Name | Importance rank | Pr ( > \|t\|) | t statistic | Description | Units | Source |
|---|---|---|---|---|---|---|
| SiteElev | 5 (5) | 0.0227 (0.001955) | -2.349 (-3.262) | Elevation of the sampling site (or locations for prediction) | m | FENZ (Leathwick et al. 2010) |
| usCatElev | 2 (2) | 4.94e-06 (6.94e-08) | 5.095(6.280) | Mean elevation of the upstream catchment | m | REC (Snelder and Biggs, 2002) |
| usAveSlope | 4 (4) | 8.50e-05 (0.000799) | -4.264(-3.562) | Mean slope of upstream catchment | ºC | REC (Snelder and Biggs, 2002) |
| usAnRainVar | 1 (3) | 6.00e-07 (8.60e-05) | 5.686(4.260) | Coefficient of variation of annual catchment rainfall | mm | FENZ (Leathwick et al. 2010) |
| usLWArea | 3 (1) | 2.79e-05 (1.27e-08) | -4.595 (-6.743) | Percentage of upstream catchment covered in lakes or wetlands | - | FENZ (Leathwick et al. 2010) |