# Peer review of "A method for predicting hydrogen and oxygen isotope distributions across a region's river network using reach-scale environmental attributes"

_Hydrology and Earth System Sciences, 2021_

## Author Response (AR3)

Responses to referee comment R1 on:

A method for predicting hydrogen and oxygen isotope distributions across a region's river network using reach-scale environmental attributes

By Bruce D. Dudley, Jing Yang, Ude Shankar and Scott Graham

**Please note that line numbers given below in the review comments (blue), and responses to comments (yellow) refer to the initial submission of the manuscript. Line numbers given in the corrections text refer to the revised manuscript 'Revised manuscript no markup_Dudley et al..pdf'**

Referee comment (RC): In this manuscript, the authors produce isoscapes for the river networks of New Zealand, based on reach-scale environmental attributes. Their data and new maps for the surface runoff isotopes could be useful contributions in the region, although there are some issues related to the contributions, data, methods and results.

Main issues

(a) The authors have to articulate their contributions clearly. They should not include irrelevant claims which take away people's attention on their real contributions of the work.

Response (R): We have interpreted this comment as a summary of the bullet points given below under the reviewer's section (a), and address these individually.

See changes below.

RC: The main contributions of this work can be (1) new isotope validation dataset (File S1; e.g. Additional monthly data for New Zealand in 2017-2020), and (2) the isotope maps of surface runoff based on precipitation isotope maps and other reach-scale environmental attributes.

Response: We agree that our stream measurements dataset, and modelled reach scale values (i.e. those available at https://shiny.niwa.co.nz/nzrivermaps/ as shown below)  may be useful contributions regionally.

However, we note that the method used to derive modelled reach scale values improves upon a method originally applied to the continental USA (Bowen et al. 2011). Our improved method is transferrable to other parts of the world and may thus be of value for an international audience. To better clarify to readers of the novelty of this method and the improvement it offers relative to other methods of mapping river water isotope values, we will improve description of the regression kriging method, its uses elsewhere, and the novelty in its application to river water isoscapes (as suggested by reviewer 2).

Change made: Text added to introduction L. 76: 'Regression kriging is spatial prediction technique often used in soil mapping that combines spatial similarity with non-location predictors (Hengl et al. 2007; Keskin and Grunwald 2018).  This approach would seem

appropriate for river water isoscapes where divergence of surface water isotope values away from those predicted using a water balance approach is partly caused by catchment characteristics that are not strongly spatially autocorrelated.'

RC: Our readers would want some more specific information related to the specific contributions of this paper on the data legacy and isoscapes in New Zealand.

Instead of just giving a summary of general processes related to rainout or temperature effects of isotopes, which has been routinely discussed in other similar previous works, the authors could provide a review of the history of environmental isotope studies over New Zealand, so that they can introduce all the crucial datasets or sampling campaigns in the country.

Response: We think a brief review of the history of environmental isotope studies in New Zealand is a good idea. We will add text to the introduction to supplement the description of the literature validation dataset in the methods (section 2.4). In response also to comments below from both reviewers, will add a table to show the sources, size and duration of datasets and how these data are used in developing, calibrating and validating the models.

Change made:

Now that I've come to write this, I've found it hard to add this information to the introduction without distracting from the main messages of the paper. Instead, I've added this information to section 2.4, which covers this subject. I hope this is acceptable. This extra text reads: ' The report of Stewart et al. (1983) includes $\delta^2H$ (but not $\delta^{18}O$) samples taken from nearly 200 sites throughout New Zealand between 1966 and 1981. These are largely single samples for rivers but include repeated sampling over several years at some sites that highlighted storm-to-storm and seasonal variation. The work of Lachniet et al. (2021) is based on a single, high spatial resolution sampling campaign across the South Island of New Zealand in 2016. Kerr et al. (2015) reported single samples taken from small rivers across an east to west transect through the Southern Alps, in the South Island. Marttila et al. (2017) conducted over two years of monthly sampling from 7 river sites in a small area of the South Island.'

Table 2 added as described.

RC: It will be good that the author can include the georeferenced maps (e.g. the GeoTIFF files) in their supplementary materials.

Response: We will include the GeoTIFF files for precipitation and river isoscapes in our supplementary materials.

Change made: GeoTIFF files for precipitation and river isoscapes now added to supplementary materials.

RC: One of the main contributions of this paper is that the authors generated surface water maps from a precipitation map. Therefore, please show the river network and catchments in Figure 1 to give people some ideas of how different isotope sampling locations can be related to their data sources or references.

Response: Figure 1 already shows the river network (panel F) and the locations of validation sites (panel G). It is an issue that at national scale, in panel F the smaller reaches and catchments blend together so that only larger rivers are visible. For this reason, panels A-F show only the Canterbury region of New Zealand. We cannot show every catchment in high detail, but a 'zoomed-in' example comparing model results for individual reaches to values from monitoring sites is presented in Figure 7.

For the reader with further interest in the river network and catchments we will add more detail as follows:

Add reference Smith and McBride (1990) to panel G in Figure 1; this reference describes the design of New Zealand's national river water quality network, and the catchments from which monthly isotope samples were taken.

Add text at the start of section 3 to give the reader better access to information about monitoring sites and their catchments, including design of the monitoring network (Smith and McBride 1990), and descriptions of physical (catchment), flow and chemical conditions at monitoring sites (Davies-Colley et al. 2011; Julian et al. 2017; Yang et al. 2020).

Change made:

Reference Smith and McBride (1990) added to panel G in Figure 1

Text added to section 2.3 (L 130): 'Modelled river water isotope values were compared to annual average values from 58 sites from the National river water quality network (NRWQN; selected to represent catchments nationally (Yang et al. 2020). Design of the NRWQN is described by Smith and McBride (1990), while descriptions of physical (catchment), flow and chemical conditions at monitoring sites can be found in Davies-Colley et al. (2011), Julian et al. (2017), and Yang et al. (2020).'

RC: Although the author used water balanced methods, I did not really see any results related to surface flow mixing or patterns. Moreover, the authors have to recognise their main contribution of the work is not about isotopes in animals or plants. Only the

implication of this work can be related to isotopes in animals or plants. However, the current abstract makes people think that the main topic of this work is about isotopes stored in animal and plant issues.

Response:

Regarding the presentation of results related to surface flow mixing or patterns:

As noted in L. 16, We used a water balance-based method to generate the river isoscape. Patterns of surface flow and mixing are therefore represented by the isoscape outputs in figures 6 and 7, in the reach scale values available at https://shiny.niwa.co.nz/nzrivermaps/ and will be available in the GeoTIFF files for river isoscapes we will add as supplementary material.

Regarding the comment about the main contribution of the work is not being about isotopes in animals or plants:

We will remove mention of animals and plants from the first sentence of the manuscript. This will read: '*Stable isotope ratio measurements (isotope values) of surface water reflect hydrological pathways, mixing processes, and atmospheric exchange within catchments*.'.

We will slightly alter the last sentence of the abstract to make it clear that we haven't measured any animals or plants. This will read: '*The resulting river water isoscapes have potential applications in ecological, hydrological and provenance studies for which understanding of spatial variation in surface water isotope values is required*.'

Change made:

As above, GeoTIFF files for precipitation and river isoscapes now added to supplementary materials.

Abstract restructured based on the comments of both reviewers. Changes include:

- Line 14 changed to 'We used a water balance-based method to predict long-term average $\delta^2$H and $\delta^{18}$O for New Zealand rivers' from 'We used a water balance-based method which represents patterns of surface flow and mixing…'
- Increased focus on water isotopes as hydrological, rather than ecological tracers.
- First sentence changed as describes in our initial response to review comments, above.
- Last sentence changed to 'The resulting isoscapes have potential applications in ecological, hydrological and provenance studies that consider differences between surface water isotope values and those of other components of the hydrological cycle (e.g. subsurface runoff or local precipitation).'

RC: In Section 3, the authors should articulate their overall results by removing irrelevant and weak discussions.

(b) The authors have to clarify the details of the data and methods. In this study, the used methods are a well-developed kriging approach. Although these used methods may not be a significant advancement for spatial analysis, they should be suitable for this manuscript's purpose. Even though it is somewhat expected, the authors showed their regression-based kriging was better than the ordinary kriging.

R: We have interpreted these comments as a summary of the bullet points given below under this review section (b), and address these individually.

Change made: See changes below.

RC: The authors recognise that that "distance-based" geospatial and statistical interpolation is less appropriate (Ln 15 and Ln 54), but their regression-based kriging methods is still "distance-based" geospatial and statistical interpolation at the end of day.

Response: We agree. We will address this by providing more background on the differences between ordinary kriging and regression kriging (as requested by reviewer 2) and using the term 'simple distance-based' to describe ordinary kriging in lines 15 and 54.

Changes made:

L. 76. Added detail to introduction on differences between ordinary and regression kriging as detailed above.

Added 'simple' in front of 'distance-based' in abstract, introduction, section 4.4 and section 3.6.

RC: In Section 2, there are not many details about how to select five environmental variables in Table 1 from Table S1 (Ln164-Ln165). There are some logic issues here. The authors used the small number of available samples to justify the use of stepwise regression to reduce the number of independent variables.

Response: We will provide more detail and support our choice of method with a reference at this point. E.g. 'From the list of independent variables in Table S1, five were selected for the regression analysis based on BIC (Baysian Information Criteria), following the "one in ten rule" (e.g. Harrell Jr (2015)), i.e. one predictive variable can be included for every ten sites in the dataset.'

We will add t values and P values to this table.

Change made:

L. 173-. Text as described in the response above.

t values and P values added to Table 2 (formerly Table 1).

RC: A table of the data for developing, calibrating and validating the models should be provided. Therefore, in the table, the authors should give the details of data sources (e.g. related publications), locations (e.g. south or north islands), sampling periods (2007-2009 in Ln 114) and number of samples (e.g. 51 sites Ln113).

Response: Really good idea. Thanks. Reviewer 2 also had trouble working out which datasets were used in which step, and what data these contained. We will add a table to make this clearer.

Change made: Table added (Table 1) giving details of isotope datasets used for calibration and validation steps. Figure 1 gives data sources used for developing the initial model.

RC: The authors should think clearly why they choose the data between 2017 and 2020 for the residual calculation (Ln 126).

Response:

We will try to make this clearer using the addition of a table as described in the previous comment.

As described in a response to reviewer 2, the 2017 to 2020 river water monitoring data were monthly samples over three years (36 samples per site) from 58 sites spread across the major catchments of New Zealand. Site mean values of $\delta^2H$ and $\delta^{18}O$ from this dataset are appropriate for correcting the river isotope model; the model gives an estimate of 'average' river water isotope values for each reach in the river network.

Other data collated from the literature used in final checking of the model (figure 5, and supplementary material S1) is largely from 'one-off' samples from river reaches, which are less appropriate for correcting the model because river water isotope values vary seasonally, and under changing flow conditions (Yang et al. 2020). These data do however provide a good independent check of how the model performs compared to other model approaches.

Change made: As above, Table added (Table 1) giving details of isotope datasets used for calibration and validation steps. The extra detail added to section 2.4 about the isotope collections in previous studies also gives some context to the use of sites with repeated sampling for correction of our model.

RC: The author mentioned a poorer longer-term fit in the other study (Ln 200). Let's think about it together here. For the annual values between 2007 and 2010, there could be only four data points for computing the correlation…

Response: We will make sure this field precipitation dataset and its use in model checking are clearly described using the additional table suggested above.

The dataset of field precipitation samples used for this correlation/model checking contained monthly values from 51 sites between 2007 and 2010. So, ca. 1400 data points for computing the monthly correlation, and 51 data points for the annual average correlation (not 4). We will add these samples size values to the manuscript text alongside the $R^2$ and RMSE values. We note that this correlation method and the field dataset are the same as used by Baisden et al. (2016). Using the same corelation method and field dataset allowed us to check our precipitation model replicated their published one well.

Change made: As above, table added (Table 1) giving details of isotope datasets used for calibration and validation steps.

RC: At the moment, the model in Equation 3 is only a first order model of environmental variables. Authors may explain why they did not try to explore higher order models for the environmental variables.

Response: We didn't apply nonlinear regression simply because it would increase the complexity for parameter estimates. This was inappropriate given limited number of sites (58) available for validation. We will make this clear in the manuscript.

Change made: text from L176 now reads 'In the second step, spatial autocorrelation is considered together with the five selected variables following Equation 3 to give the prediction. Similarly, since non-linear regression requires an increased number of coefficients to be estimated we also used linear regression in Equation 3 to avoid overfitting.'

RC: In Section 3, the authors should try to discuss how their selected environmental variables can be related to ground water and vegetative surface (Ln49-Ln50). The author did recognise that their model system was biased (Ln 403) which is very likely related to their selected environmental variables in Table 1.

Response: We will revise this section for clarity.

Spatial patterns of residuals in our method, and predictors (e.g. those in Table 1) could be used to increase understanding of hydrological processes. A simple example of this is that upstream wetland and lake area, which leads to higher evaporative fractionation and thus higher river water $\delta^2H$ and $\delta^{18}O$ values, explained spatial patterns of residuals in our study,

which used the (Bowen et al. 2011) water balance model that assumes no evaporative fractionation.

A similar approach may be taken for lowland reaches gaining a large portion of their flow from high-elevation-derived groundwater. These reaches may show up as more isotopically negative than would be expected based on recharge and surface routing of local, low-elevation rainfall.

HOWEVER – our manuscript shows that this type of approach, and similar approaches in isotope enabled hydrological models (e.g. Belachew et al. (2016)) are reliant on the accuracy of the precipitation isotope model. We believe that some of the variables in table 1 reflect correction of spatial inaccuracies of the precipitation model.

We will adjust the discussion accordingly.

Change made: Text from L. 418 now reads: 'Improved regional precipitation isotope input data would raise the visibility of hydrological fluxes (such as high-elevation-derived groundwater contributions to rivers) in the regression kriging correction steps of our method. Similarly, improved regional precipitation isotope models are likely to improve the performance of process-based isotope hydrology models, such as the Isotope-enabled coupled catchment–lake water balance model (Belachew et al., 2016), that are designed to quantify hydrological fluxes (e.g. between rivers, lakes and groundwater) using water isotope data.'

RC: In Equation 1, there is no storage consideration. In the implication section, the authors should discuss how storage can affect their overall map results in Section 3.

Response: Our method is fairly robust in this respect. Because both input data and data used to correct the model are averaged to give 'steady state' values, seasonal variation in contributions of surface and groundwater flows (which may have differing isotope values) to rivers is incorporated.

We will add brief discussion on this point.

Change made: Text from L. 336 now reads 'The most appropriate sites for validation of our model were those 23 sites for which long-term monthly sampling records are available, enabling us to compare predicted and measured annual average values. Average values derived from monthly measurements made over several years are likely to 'average out' much of the temporal variation in river water isotope values which results from temporal variation in precipitation isotope values, evaporation, and contributions from different flow pathways. At these sites, the model explained 90.6 % of the $\delta^2$H variance across the dataset.'

RC: (c) Some interpretation of results can be problematic and speculative. More discussion of the limitations of the study is needed.

Response: We have interpreted this comment as a summary of the specific bullet points given below under this review section (c), and address these individually.

Change made: Changes made detailed below.

RC: In L260-L285, the discussions and interpretations related to air masses, regional circulations and orographic effects are very speculating. These discussions are without much strong quantitative evidence in the manuscript.

Response: We agree that the discussion of the effects of origin of air masses on precipitation isotope values currently looks speculative because we haven't referenced previous work well enough. We will add references to studies of isotopes in precipitation in New Zealand (e.g. (McDonnell 1988) to back up this point.

We feel our discussion of regional orographic effects is well supported by the work of Purdie et al. (2010) and Kerr et al. (2015), which we have referenced in the manuscript. We have gone to further effort to back up our statements using Appendix Figure 1 and its accompanying text. However, we will add international references showing the same effects in other mountainous regions worldwide to support our statements.

Changes made: We have cut back the section that was in L260-L285 of the original manuscript (in the middle of section 3.4) to reduce repetition between sections in the next reviewer comment, and have added references (Stewart et al. 1983) and (McDonnell 1988) to support our interpretation related to air masses, regional circulations and orographic effects in L 418-429.

RC: For example, the results in L223-L235 are very hypothetical. They are also very repetitive in the manuscript, because the authors repeat these speculations again in Section 3.4. Moreover, the current results are only marginally or speculatively related to cloud processes in Ln43.

Response: Firstly, we will directly reference Figure 2 in the statement on L. 225-226; i.e. 'of the eight sites where predicted $\delta^{18}O$ values exceed average measured $\delta^{18}O$ values by > 1‰ (Figure 2), seven are in alpine-fed rivers on the leeward east of New Zealand.'

We agree that there is some unnecessary repetition between sections 3.2 and 3.4. We will work to reduce or remove this.

We will revise the discussion to improve the description of links between:

1.      The predictors of residuals in Table 1

2.      The spatial inaccuracy of the precipitation model (and its likely causes)

3.      Our ability to improve understanding of processes in hydrology using our approach.

Put simply, Table 1 currently contains predictors that correct for spatial inaccuracy of the precipitation model. If we can improve the precipitation model, our method will be more useful for understanding of processes in hydrology (such as evaporation and groundwater contributions to surface water) that change in isotope values of river water.

Changes made:

Reference to Figure 2 added to L. 242.

Added reference to Stewart et al. (1983) to give more weight to the assertion on L. 244 that Rivers and streams leeward of the Southern Alps show isotopically depleted values characteristic of spillover of orographic precipitation from the windward west of the Southern Alps.

We have cut back the description of orographic effects on precipitation in section 3.4 to reduce repetition.

We have revised the discussion and added text to L411-428 following the three points laid out in our response above.

RC: The authors should revise their discussion, similar to Ln 285-L302 where the authors discussed their result based on the fitted model variable results (e.g. usAnRainVar).

Response: As above, we will revise the discussion to improve the description of links between:

1.      The predictors of residuals in Table 1

2.      The spatial inaccuracy of the precipitation model (and its likely causes)

3.      Our ability to improve understanding of processes in hydrology using our approach.

Change made: Changes made to L 295-301 as shown in the response to the next comment, and to L418-428 as described in the response to the previous comment.

RC: For orographic effects, the authors may need to consider more about "aspect" and "wind" variables in their models, so that they can justify their discussion based on Kerr et al. (2015).

Response: Good point. In fact, the 'usAnRainVar' variable in Table 1 is strongly correlated with aspect. We will make this clearer as described above.

Change made: L. 297-300 now reads 'All predictors in Table 2 except upstream lake and wetland area show a strong west to east gradient; for example, areas to the east of the Southern Alps where the water balance model overpredicts river water $\delta^2$H and $\delta^{18}$O values have lower average catchment slopes and higher upstream annual rainfall variability than are present to the west the alps where the water balance model underpredicts river water $\delta^2$H and $\delta^{18}$O values'

RC: As I have mentioned in my first comments, the results of this work are unlikely to be useful for studying movement of aquatic organisms (L430). The current maps are only for hydrogen and oxygen. There were no other isotope results such as nitrogen. In general, the discussion of animal and plant tissues (Ln10) is far-fetching in this manuscript. The results of this paper are not really giving much insights into them.

R: The reviewer is right that the current maps have potential use in hydrological studies. With the reviewer's comments and the readership of HESS in mind, we will make modifications to the abstract, introduction and discussion to lessen the focus on ecological implications of this work and increase focus on hydrological uses and implications.

We do not feel that the absence of nitrogen data from our paper negates the usefulness of our work to (for example) ecological research. While having MORE tracers is almost always better in mixing models (Fry 2006), hydrogen and oxygen stable isotopes are useful nonetheless for aquatic ecology (Soto et al. 2013).

We do not agree that the results of this work are unlikely to be useful for the ecological purposes we have outlined in our manuscript. The geographical distributions of hydrogen and oxygen isotopes in precipitation and surface water form underpin a rich and growing body of research into animal migrations, as well as other cross-disciplinary uses. Quoting from Bowen et al. (2009) 'Isoscapes have great power as a cross-disciplinary research tool, as exemplified by the translation of hydrology-focused GNIP [Global Network of Isotopes in Precipitation] data into tools for animal migration research.'. Examples of ecological (migration) research based on GNIP hydrogen and oxygen isotope data are included in a review by Hobson and Wassenaar (2018). The Global Network for Isotopes in Rivers (GNIR) has similar aims. Quoting from Halder et al. (2015) 'The aim of the GNIR programme is to collect and disseminate time-series and synoptic collections of riverine isotope data from the world's rivers and to inform a range of scientific disciplines including hydrology,

meteorology and climatology, oceanography, limnology, and aquatic ecology.' However, the reviewer's comments make it plain that we have not conveyed this potential for cross-disciplinary use of our work adequately. To address this, we will add brief but specific examples to the manuscript on this topic to section 4.

Changes made:

L. 46- We have added a couple of examples to the introduction to give hydrological and ecological applications of precipitation isoscapes. This reads: 'Understanding of the processes controlling precipitation isotope values has aided development of precipitation and surface water isotope maps (Bowen et al. 2011; Bowen and Revenaugh 2003), with resulting hydrological and cross-disciplinary applications (Jasechko et al. 2013; Vander Zanden et al. 2016).'

We have reworked the conclusions section from L. 434 and added an example of use of $\delta^{18}O$ values in anthropological studies. We have added the Soto et al. 2013 reference to this conclusions section to provide the reader with examples of water isotope uses in aquatic ecology.

RC: The system bias of this study (L403) is unlikely to help others improve understanding of isotope patterns. Therefore, the authors should try to reframe their writing by reducing their discussion based on speculations, and suggest more how we can improve our understanding patterns of precipitation isotope values by using hydrological process-based models to investigate how flow and evaporation processes affect isotope patterns.

R: We will extend the focus of this paragraph outside of the scope of the current study, towards more general discussion of using river isotope models to understand hydrological processes.

We will restructure this paragraph as follows:

1. Some isotope-enabled hydrological models (e.g. Belachew et al. (2016)) use precipitation isotope models as input data to give improved estimates of fluxes between components of the hydrological cycle.

2. The accuracy of these flux estimates relies partly on accuracy in input data from precipitation isotope models

3. Data from precipitation isotope models will always be imperfect, but improvements in the accuracy of precipitation isotope models can improve our understanding of flow pathways and evaporation processes at landscape scales.

Change made: Paragraph beginning L. 414 changed as described.

RC: Currently, I did not see much mixing and surface flow results which is suggested in Ln16. I also did not see the dam results mentioned in Ln13 and Ln68.

Response:

We will add t values and p values to Table 1 to better support the discussion around upstream lake and wetland area effects on $\delta^2H$ and $\delta^{18}O$ values of river water. We will refer the reader to these results in section 3.3.

Open water behind dams is included in the variable usLWArea. We will add a specific reference to dams to section 3.6.

For clarity, we will replace abbreviated variable names (e.g. 'usLWArea') in the results and discussion text with full variable names (e.g. 'Upstream lake and wetland area').

Line 16 states 'We used a water balance-based method, which represents patterns of surface flow and mixing'. Thus, mixing results are incorporated into the water balance results shown in (for example) Figure 2, 6 and 7, and in the online maps shown below.

Change made: t values and p values to Table 2 (formerly Table 1).

L. 368 – 372 now reads: The effect of lakes and wetlands is an example of how landscape processes control $\delta^2H$ and $\delta^{18}O$ values in rivers; three factors combine to make regression kriging a particularly appropriate method to represent these processes. Firstly, lakes (including artificial lakes and dams) do not cluster predictably across New Zealand, and secondly, their effects are confined to downstream reaches. Thus, a modelling approach that considers the dendritic nature of river networks in this correction step is likely to better account for this process than one which corrects based only on Euclidean distance (Brennan et al., 2016).

Abbreviated variable names now described in full in section 3.3. (e.g. see L. 263-275.)

RC: Until the authors could have results similar to Figure 7 for all the main catchments in New Zealand, the discussion in Ln355 - Ln379 could not be justified. For example, there are no similar results of Figure 7 for the South Island in the manuscript.

Response:

We will add t values and p values to Table 1 to better support this discussion section.

Discussion of relationships between environmental variables and river water $\delta^2H$ and $\delta^{18}O$ in Ln355 - Ln379 is backed up by multivariable regression results. Importance ranks for this regression for $\delta^2H$ and $\delta^{18}O$ residuals are already given in table 1, based on t statistics. This

regression used $\delta^2$H and $\delta^{18}$O data from across New Zealand, not just the catchment in figure 7. Our discussion on Ln355 - Ln379 is limited to statistically significant predictors shown in table 1, of which upsteam lake and wetland area is one.

We could produce plots similar to Figure 7 for all the main catchments in New Zealand, but it is not practical to show them all in the manuscript. Figure 7 gives an example. We have provided access to data shown in Figure 7 at https://shiny.niwa.co.nz/nzrivermaps/. A South Island example is shown below:

[Figure]

Monitoring data (i.e. equivalent to the points in Figure 7, but across major catchments nationally) are available via the IAEA WISER portal.

Change made: t values and p values added to Table 2 (formerly Table 1).

RC: Perhaps, the authors can have more discussion on how results in Figure 7 are related to the "dendritic" patterns (Ln62).

Response: Certainly. We will add some detail on this to the section in L. 364-378.

Change made: This topic is covered on L. 364 – 392, and we have added the following to L. 381 for clarity: 'Using regression kriging correction, the residual correction follows the 'dendritic' nature of the river system and is passed only to downstream reaches.'.

RC: More insightful thoughts on variations between precipitation and surface water will be useful to demonstrate the values of this work. It would be great to have more quantification

and discussion on how the precipitation and new runoff maps could be different in terms of their patterns.

Response: Really good point. We will add more discussion to section 4, focussing on implications of differences in isotopes in precipitation and those shown in our runoff maps. In terms of quantification, to some degree this is already visible in the isotopic differences between rivers fed by high elevation recharge and those fed by local lowland recharge (see above). We will add some text to this effect.

Change made: We have now focussed on this topic in the final conclusions and implications section from L. 440 as follows '*Distinct differences in spatial patterns of $\delta^2H$ and $\delta^{18}O$ in river waters to those of precipitation highlight the value of river isoscapes in cross-disciplinary research. Differences between precipitation and river water isotope values were particularly evident at low elevations; New Zealand's high central mountain ranges create strong elevation gradients in precipitation isotopes, and lowland regions receiving more isotopically enriched rainfall are intersected by alpine-fed rivers bearing isotopically depleted water from high-elevations. In addition to the hydrological implications described above, quantification of isotopic values for water sources across elevation gradients may be of particular benefit to those studies that attempt to attribute organic material (such as sediment-bound organic material transported in rivers) to particular elevation bands (Feakins et al. 2016; Upadhayay et al. 2017). The river water isoscapes shown in this study are also likely to be appropriate for studies utilising $\delta^{18}O$ values in human tissues to determine historical migration patterns (e.g. King et al. (2021)); local drinking water.*' provides a good proxy for the majority of total water intake in humans (Ehleringer et al. 2008; Guelinckx et al. 2016).

As requested, we have included the geotiff maps of precipitation and river water isoscapes as supplement material to aid quantification of differences between these.

RC: Overall, the data of this work could be useful regionally.

Thank you. As above, to better clarify to readers the international transferability of our work, we will improve description of the regression kriging method, its uses elsewhere, and the novelty in its application to mapping river water isotope values (as also suggested by reviewer 2).

Change made: Additional text on regression kriging added to introduction as described above.

**References:**

Alexander, R.B., A.H. Elliott, U. Shankar, and G.B. McBride. 2002. Estimating the sources and transport of nutrients in the Waikato River Basin, New Zealand. *Water Resources Research* 38: 4-1-4-23.

Bowen, G.J., C.D. Kennedy, Z. Liu, and J. Stalker. 2011. Water balance model for mean annual hydrogen and oxygen isotope distributions in surface waters of the contiguous United States. *Journal of Geophysical Research: Biogeosciences* 116.

Bowen, G.J., and J. Revenaugh. 2003. Interpolating the isotopic composition of modern meteoric precipitation. *Water Resources Research* 39.

Craig, H. 1961. Isotopic variations in meteoric waters. *Science* 133: 1702-1703.

Davies-Colley, R.J., D.G. Smith, R.C. Ward, G.G. Bryers, G.B. McBride, J.M. Quinn, and M.R. Scarsbrook. 2011. Twenty Years of New Zealand's National Rivers Water Quality Network: Benefits of Careful Design and Consistent Operation1. *JAWRA Journal of the American Water Resources Association* 47: 750-771.

Ehleringer, J.R., G.J. Bowen, L.A. Chesson, A.G. West, D.W. Podlesak, and T.E. Cerling. 2008. Hydrogen and oxygen isotope ratios in human hair are related to geography. *Proceedings of the National Academy of Sciences* 105: 2788-2793.

Elliott, A.H., R.B. Alexander, G.E. Schwarz, U. Shankar, J.P.S. Sukias, and G.B. McBride. 2005. Estimation of Nutrient Sources and Transport for New Zealand Using the Hybrid Mechanistic-statistical Model SPARROW. *Journal of Hydrology (New Zealand)* 44: 1-27.

Feakins, S.J., L.P. Bentley, N. Salinas, A. Shenkin, B. Blonder, G.R. Goldsmith, C. Ponton, L.J. Arvin, M.S. Wu, and T. Peters. 2016. Plant leaf wax biomarkers capture gradients in hydrogen isotopes of precipitation from the Andes and Amazon. *Geochimica et Cosmochimica Acta* 182: 155-172.

Gat, J.R. 1996. Oxygen and hydrogen isotopes in the hydrologic cycle. *Annual Review of Earth and Planetary Sciences* 24: 225-262.

Guelinckx, I., G. Tavoularis, J. König, C. Morin, H. Gharbi, and J. Gandy. 2016. Contribution of water from food and fluids to total water intake: analysis of a French and UK population surveys. *Nutrients* 8: 630.

Hengl, T., G.B.M. Heuvelink, and D.G. Rossiter. 2007. About regression-kriging: From equations to case studies. *Computers & Geosciences* 33: 1301-1315.

Jasechko, S., Z.D. Sharp, J.J. Gibson, S.J. Birks, Y. Yi, and P.J. Fawcett. 2013. Terrestrial water fluxes dominated by transpiration. *Nature* 496: 347.

Julian, J.P., K.M. de Beurs, B. Owsley, R.J. Davies-Colley, and A.G.E. Ausseil. 2017. River water quality changes in New Zealand over 26 years: response to land use intensity. *Hydrol. Earth Syst. Sci.* 21: 1149-1171.

Keskin, H., and S. Grunwald. 2018. Regression kriging as a workhorse in the digital soil mapper's toolbox. *Geoderma* 326: 22-41.

King, C.L., H.R. Buckley, P. Petchey, P. Roberts, J. Zech, R. Kinaston, C. Collins, O. Kardailsky, E. Matisoo-Smith, and G. Nowell. 2021. An isotopic and genetic study of multi-cultural colonial New Zealand. *Journal of Archaeological Science* 128: 105337.

McDonnell, J.J. 1988. The age, origin and pathway of subsurface stormflow in a steep humid headwater catchment. PhD thesis, University of Canterbury Canterbury, New Zealand.

Smith, D.G., and G.B. McBride. 1990. New Zealand's national water quality monitoring network - design and first year's operation. *JAWRA Journal of the American Water Resources Association* 26: 767-775.

Stewart, M.K., M.A. Cox, M.R. James, and G.L. Lyon. 1983. Deuterium in New Zealand rivers and streams, 42. Lower Hutt: DSIR, Institute of Nuclear Sciences: Institute of Nuclear Sciences

Upadhayay, H.R., S. Bodé, M. Griepentrog, D. Huygens, R.M. Bajracharya, W.H. Blake, G. Dercon, L. Mabit, M. Gibbs, B.X. Semmens, B.C. Stock, W. Cornelis, and P. Boeckx. 2017. Methodological perspectives on the application of compound-specific stable isotope fingerprinting for sediment source apportionment. *Journal of Soils and Sediments* 17: 1537-1553.

Vander Zanden, H.B., D.X. Soto, G.J. Bowen, and K.A. Hobson. 2016. Expanding the isotopic toolbox: applications of hydrogen and oxygen stable isotope ratios to food web studies. *Frontiers in Ecology and Evolution* 4: 20.

Yang, J., B.D. Dudley, K. Montgomery, and W. Hodgetts. 2020. Characterizing spatial and temporal variation in $^{18}O$ and $^2H$ content of New Zealand river water for better understanding of hydrologic processes. *Hydrological Processes*.

Responses to referee comment R2 on:

A method for predicting hydrogen and oxygen isotope distributions across a region's river network using reach-scale environmental attributes

By Bruce D. Dudley, Jing Yang, Ude Shankar and Scott Graham

RC: The paper introduces a new method for predicting isotope distribution using information on the river network and environmental variables. The method is applied to NZ using a number of existing databases and some extra data collected by the research team.

The approach is novel and has potential for the improvement of the predictions, although the actual application to the NZ situation does not show a striking improvement over a more traditional method. I believe this fact should be reflected in the abstract and the conclusions more clearly, so the reader does not have excessive expectations.

Response: We agree that the majority of the variation in river water isotope values across New Zealand can be explained by the water balance model used by Bowen et al. (2011) for the continental USA. This is a good point, and we will change the abstract and conclusions to make it clear.

Nevertheless, the additional effort put into regression kriging does result in improved predictions, and there are some areas of river networks, such as downstream of dams and wetlands, where it appears particularly beneficial.

Change made: Text added to abstract L. 18: 'Much of the spatial variability in $\delta^2$H and $\delta^{18}$O of New Zealand river water was explained using the initial combination of a precipitation isoscape and simple water balance model. $\delta^2$H and $\delta^{18}$O isoscapes produced by subsequently applying residuals from the water balance model as a correction factor across the river network using regression kriging showed improved fits to validation data, compared to correction using ordinary kriging.'

This point is also covered from L. 351, which reads: 'Across New Zealand, spatial and temporal patterns of $\delta^2$H and $\delta^{18}$O in precipitation and runoff are dominant drivers of $\delta^2$H and $\delta^{18}$O in river water (Figure 6). The (uncorrected) water balance model, which explicitly represents these factors, explained much of the variance present in our long-term river water dataset.'

RC: I believe a part of the methodology that is somehow understated by the authors is the use of the regression Kriging technique. I would highlight this more throughout the paper and give a bit more background in the introduction and discussion about its rationale, implementation and potential. If this is a common tool used elsewhere discuss its novelty in the application of this particular problem.

Response: Yes, regression kriging is tool used elsewhere, particularly in soil mapping (e.g. see (Hengl et al. 2007; Keskin and Grunwald 2018). We will add text to the introduction to describe other common uses of regression kriging and discuss its novelty (and appropriateness) in its application to this particular problem.

Change made: Text added to introduction L. 76: 'Regression kriging is spatial prediction technique often used in soil mapping that combines spatial similarity with non-location predictors (Hengl et al. 2007; Keskin and Grunwald 2018).  This approach would seem appropriate for river water isoscapes where divergence of surface water isotope values away from those predicted using a water balance approach is partly caused by catchment characteristics that are not strongly spatially autocorrelated.'

RC: The period of analysis of the paper is rather short, 2017-2020. An acknowledgement of this fact and the justification for why it has not been possible to use an extended period would be great. Also, what are the expectations into the future when more data becomes available?

Response: The period of analysis was limited by availability of field (monitoring) data. We will add text to the manuscript to make this clear.

Because the water balance model gives an estimate of flow-weighted average $\delta^2H$ and $\delta^{18}O$ at any reach, we needed a comparable average at each of our 58 river monitoring sites. While more data is always better, $\delta^2H$ and $\delta^{18}O$ values at our monitoring sites were relatively consistent across the three years of monitoring data we have, so we are confident that our 2017-2020 dataset is adequate for the purposes of model correction. As a rough illustration of this consistency between years, the isotope biplots below show all monthly river water samples from our 58 validation sites, with additional years of data showing in red, then blue. We note also that the monitoring dataset we have collected is the largest and by far the longest record of river water isotopes available for New Zealand.

[Figure]

River water monitoring network results 2017-2020

In response to the reviewer's question about expectations for the future: we will make our code publicly accessible so that updates to modelling methods, and additional data can iteratively improve these maps. We continue river sampling at monitoring sites, and we are also now working to improve the national precipitation isotope model. These new input and validation data will be incorporated into revised maps. When available, updated maps will be made available online via https://shiny.niwa.co.nz/nzrivermaps/

Additional (measured) river data will be added to the dataset we have provided to the IAEA WISER database.

Change made:

Added description of validation river dataset to Table 2.

Year 3 and 4 river data will be added to IAEA WISER database when year 4 data available.

We are still working on getting our code in a state suitable to be made publicly available – this is being done along with the revision of the precipitation model.

RC: Minor comment: Line 111: "Hence, we checked the results of our procedure by performing regressions between our modelled, amount-weighted monthly precipitation isotope values and measured values from the dataset of Baisden et al. (2016), comprising monthly collections from 51 sites across New Zealand between 2007 and 2009." Why is this check carried on over only two years? Again, a justification here would help the reader understand a bit better the limitations of the study.

Response: This period of analysis was again limited by availability of field (measured) data. We will add text to the manuscript to make this clear. The reviewer has also highlighted a typo – the dataset of Baisden et al. (2016) extended into early 2010 at some sites. We will correct this.

Change made: Added description of precipitation dataset to Table 2, with correct period. Corrected text on L. 119 '…51 sites across New Zealand between 2007 and 2010'

**Second round of review and responses:**

Responses to referee comment R1 on:

A method for predicting hydrogen and oxygen isotope distributions across a region's river network using reach-scale environmental attributes

By Bruce D. Dudley, Jing Yang, Ude Shankar and Scott Graham

Referee comment (RC):

Generally, the authors have addressed my comments well. The overall narrative of the paper has improved. The authors also demonstrated how the final regression kriging is better than ordinary kriging because of environmental variables. If the authors can address the following comments, this work could be publishable.

My main remaining comment is how monthly isotope data from 58 sites for 3 years can be justified in producing representable maps for 600,000 reaches and over 400,000 kilometres of rivers. When the authors have only 58 sites for 600,000 reaches, we will always have a question of how much we can trust the maps generated from this study. Nevertheless, the authors added t values and P values in Table 2 to provide us with some statistics to illustrate the usefulness of five selected environmental variables and give us some confidence in their maps.

To further see how these five environmental variables for these 58 locations can be represented for 6000,000 reaches, the authors should provide scatter plots between hydrogen & oxygen isotopes and five environmental variables, so that we can see these empirical relationships qualitatively. I would expect that some scatterplots would have poor linear relationships or highly clustered data points (e.g. isotopes vs SiteElev). However, I would like to see these plots presented frankly.

Response (R):

We appreciate the reviewer's general concern that 58 sites cannot represent an entire river network.  However, this is the essence of our water balance-based regression kriging approach.  As we note in the manuscript, a simple kriging of sampled values may give poor predictions. However, by accounting for spatial variation in precipitation isotopes and flowpaths using the water balance model, then (in the regression correction step) including well known drivers of other processes contributing to variation in river water isotopes (such as isotopic fractionation), we can extrapolate our results more widely.  We have added further references supporting this approach.

We note that we have produced scatter plots between hydrogen & oxygen isotopes at the 58 sites used in our study and environmental variables for a previous paper: Figure 6 of Yang et al. (2020) - below.

[Figure]

**FIGURE 6** Relationships between catchment environmental factors and $\delta^2$H of river water at all NZRWQN sites

Indeed, these relationships helped to inform our approach in this manuscript, as we have described on L 167-171. However, we think there would be little benefit in reproducing something like these in the current manuscript either to support the accuracy of our maps, or aid interpretation of our results for the following reasons:

**Regarding the accuracy of our maps:**

1. Figure 5 in our current manuscript gives fit statistics for linear regressions between $\delta^2$H predictions from our model and hundreds of independent data points from among the 600,000 reaches of the NZ river network. These independent $\delta^2$H measurements are not from the 58 sites sampled (for 36 months) for model correction. We used these independent $\delta^2$H measurements in Figure 5 of the current manuscript to quantify the performance of the model. $\delta^{18}$O fits are reported in the manuscript text.

[Figure]

Figure 5.

2. Panels C and D of Figure 5 show that the residual-corrected model used to make final maps gives:

    a. An improvement over the uncorrected model (Panel A) and:

    b. A good fit to literature data ($R^2$ = 0.91 and RMSE = 2.99‰ for $\delta^2$H when compared to independent long-term monitoring data (Figure 5D)). For comparison, the final model of Bowen et al. (2011) had a RMSE for $\delta^2$H of 9.2‰ when compared to long-term monitoring data - equivalent to our Figure 5D.

**With regards to interpreting drivers of residuals:**

1. Table 2 already gives t values and P values for regressions between isotope residuals and environmental variables.

2. From panel A of Figure 5 we can see that most of the variance in $\delta^2$H values of river water was explained by the combination of a precipitation isoscape and a simple water balance model. This point was noted by reviewer 2 in their first review and highlights how important the precipitation model is for river water model accuracy.

3. Currently, we think much of the residual correction using the 5 environmental variables in Table 2, and isotope data from our 58 monitoring sites is correcting for errors in the precipitation model. We explain this from L. 419 onwards. We do not think it is worthwhile to present plots of these regressions in the manuscript because while they would be very interesting with an accurate precip. model they are less interesting hydrologically if the precipitation model is inaccurate.

Changes made:

- Added labels A, B, C, D to Figure 5. These were omitted in error and might have made Figure 5 hard for the reviewer to interpret.

- Text added to L. 357 '…*and see Yang et al. (2020).*'
- Text added to L. 135: '*Measurements from this network have been used to develop and calibrate a range of hydrological and water quality models (e.g. Alexander et al. (2002), (Elliott et al. 2005)).*'

**Minor comments**

(M1) In Line 23, please state clearly what "additional hydrological processes" are.

R: Yes, good idea.

Change made: Sentence changed to '*Hence, additional hydrological process information such as evaporation effects can be incorporated into river isoscapes using regression kriging of residuals.*'

(M2) Please explain why the important ranks of environmental factors in Table 2 for oxygen and hydrogen isotopes differ, using some explanations based on New Zealand's physical environments.

R: We can certainly speculate, but our analysis does not allow us to say for certain.

Change made: Text added to L. 264: '*A possible cause for the higher ranking of upstream lake and wetland area in the $\delta^{18}O$ regression is the greater sensitivity of the $^{18}O$ component of water to kinetic fractionation effects than the $^2H$ component (Craig 1961; Gat 1996).*'

(M3) The authors want their isoscapes to be used for hydrological studies (Line 25). It would be useful if the authors could have regression kriging of four environmental variables (i.e. SiteElev, usCatElev, usAveSlope and ust.WArea) for Figures 4, 6 and 7. In hydrological studies, precipitation variations are commonly used. Regression kriging models based on four environmental variables without using precipitation as a dependent variable will be more useful for hydrological studies based the water budget.

R: Our understanding is that the reviewer is asking for us to remove the top predictor from our residuals regression (usAnRainVar, Table 2) and reanalyse without it.

We'd prefer to keep the current regression structure (i.e. 5 environmental variables) for the following reasons:

1. Removing the top predictor from our residuals regression will make our maps less accurate.

2. We think that the main benefits of our work rely on accurate maps of river water isotope values that will allow hydrologists (and others) to identify useful isotope gradients; for example, differences between local precipitation/recharge, groundwater and river water.

Change made: No change made.

(M4) It is great that the authors provide their information on https://shiny.niwa.co.nz/nzrivermaps/. The problem is that https://shiny.niwa.co.nz/nzrivermaps/ is very bulky and it is not easy to use.

At the moment, I could not produce a plot like Figure 7 that includes gauging sites, from https://shiny.niwa.co.nz/nzrivermaps/

The authors should provide a note of how to use https://shiny.niwa.co.nz/nzrivermaps/ to generate Figure 7. If the authors use R to generate their maps, they can provide their code and data.

R: We have now provided careful instructions on how to visualise and download our model data using nzrivermaps. These are in supplementary file S3. In the same file we have also provided

instructions on how to compare these nzrivermaps data to measured data from NRWQN sites, and environmental classes across the river network (see below).

Many different applications can simply be used to make maps using the data we have provided. We used a mix of ARCGIS and R and we don't think providing our R mapping code would help the reader much. We have recommended the use of ARCGIS in supplementary file S3.

Changes made:

- Supplementary file S3 added
- Text added to 'Code and data availability' section '*Instructions for accessing and comparing datasets used in this work are provided in supplementary file S3.*'

(M5) From https://shiny.niwa.co.nz/nzrivermaps/, we know that there are different climate classes, geology classes, landcover classes, Strahler stream orders, valley landform classes and topographical classes. Please provide a table to show how the 58 NRWQN stations and 600,000 reaches are distributed in these classes to let our readers know how these 58 NRWQN stations represent 600,000 reaches.

R: This network of sites was designed to be representative of New Zealand River environments, to facilitate analyses of the type performed in this study. We have already directed the reader to the information the reviewer is requesting - on L. 130, which reads '*Modelled river water isotope values were compared to annual average values from 58 sites from the National river water quality network (NRWQN; selected to represent catchments nationally (Yang et al. 2020)). Design of the NRWQN is described by Smith and McBride (1990), while descriptions of physical (catchment), flow and chemical conditions at monitoring sites can be found in Davies-Colley et al. (2011), Julian et al. (2017), and Yang et al. (2020).*'

We also now note that other studies that use data from these sites to calibrate large-spatial-scale water chemistry models include Alexander et al. (2002) and (Elliott et al. 2005).

We have added a table of site information to Supplementary file S3, as shown in the screenshot below. This includes the river segment identifier that allows the reader to compare data from the 58 NRWQN sites with information from the entire River Environment Classification (REC) database including all of the classes the reviewer mentions, and many others.

[Figure]

which is available from the NIWA website (https://niwa.co.nz/freshwater-and-estuaries/management-tools/river-environment-classification-0). The joining column in the two datasets is labelled *nzsegment*.

**Comparison with point measurements of river water isotopes**

Table 1 provides site information for National River Water Quality Network sites from which stable isotope data has been collected since 2017. These isotope data are stored online through the IAEA GNIR programme. It can be downloaded from the WISER database at https://nucleus.iaea.org/wiser

This data can used with modelled isotope values and geographical predictor information by combining it with New Zealand digital river network and NZ River Maps data using the *nzsegment* joining column.

**Table 1. Site information for NRWQN isotope sampling sites.**

| Site Code | river | Catch. Area km2 | Highest Catch. Elev (m) | Site Elev (m) | lat | long | nzsegment |
|---|---|---|---|---|---|---|---|
| AK1 | Hoteo | 270 | 107 | 15 | -36.3862 | 174.5112 | 2001653 |
| AK2 | Rangitopuni | 82 | 228 | 10 | -36.7349 | 174.6182 | 2004545 |
| AX1 | Clutha | 4453 | 973 | 305 | -44.7328 | 169.2802 | 14014867 |
| AX2 | Kawarau | 4302 | 1043 | 305 | -45.0093 | 168.8785 | 14027448 |
| AX3 | Shotover | 1079 | 1200 | 320 | -44.9918 | 168.7163 | 14026862 |
| AX4 | Clutha | 16548 | 902 | 91 | -45.6632 | 169.4057 | 14055045 |
| CH1 | Hurunui | 1060 | 976 | 442 | -42.7922 | 172.543 | 13020391 |
| CH2 | Hurunui | 2525 | 648 | 60 | -42.902 | 173.1009 | 13023957 |
| CH3 | Waimakariri | 2387 | 1034 | 244 | -43.3621 | 172.0557 | 13040507 |
| CH4 | Waimakariri | 3076 | 854 | 76 | -43.423 | 172.634 | 13042388 |
| DN2 | Sutton Stm | 151 | 672 | 220 | -45.5979 | 170.094 | 14052240 |
| DN4 | Clutha | 20582 | 790 | 9 | -46.2384 | 169.746 | 14070057 |
| DN5 | Mataura | 5139 | 470 | 15 | -46.3877 | 168.7914 | 15059190 |
| DN7 | Oreti | 1139 | 694 | 220 | -45.7186 | 168.4308 | 15033324 |
| GS1 | Waipaoa | 1571 | 385 | 55 | -38.4684 | 177.8777 | 5010343 |
| GS2 | Waikohu | 30.5 | 722 | 457 | -38.4172 | 177.5601 | 5009160 |
| GS3 | Motu | 293 | 622 | 425 | -38.2024 | 177.6195 | 4016696 |
| GS4 | Motu | 1376 | 607 | 11 | -37.8616 | 177.636 | 4005116 |
| GY1 | Buller | 6309 | 736 | 15 | -41.8344 | 171.7013 | 12012463 |
| GY2 | Grey | 3827 | 485 | 20 | -42.4531 | 171.2992 | 12028095 |
| GY3 | Grey | 642 | 774 | 171 | -42.3616 | 171.7842 | 12025991 |
| GY4 | Haast | 1027 | 1009 | 53 | -43.9445 | 169.2987 | 12052272 |
| HM1 | Waipa | 304 | 413 | 80 | -38.2692 | 175.3501 | 3029370 |
| HM2 | Waipa | 2822 | 201 | 10 | -37.7992 | 175.1492 | 3017829 |
| HM6 | Ohinemuri | 305 | 248 | 10 | -37.417 | 175.7155 | 3010506 |
| HV2 | Tukituki | 2438 | 342 | 26 | -39.7164 | 176.9285 | 8026822 |
| HV3 | Ngaruroro | 2001 | 663 | 2 | -39.5879 | 176.8877 | 8024658 |

**Download data from NZ River Maps**

NZ River Maps offers the option of downloading data for use in other applications under the Download data tab. The data provided in NZ River Maps is made available to download free of charge to allow you to use it for your own research. All data unless specifically stated is licensed under a Creative Commons Attribution 3.0 New Zealand License and must be attributed back to its original creator. We ask that you also acknowledge the use of NZ River Maps using the suggested citation.

Whitehead, A.L., Booker, D.J. (2020). NZ River Maps: An interactive online tool for mapping predicted freshwater variables across New Zealand. NIWA, Christchurch. https://shiny.niwa.co.nz/nzrivermaps/

To download data:

1. Select your desired metrics. All predictions within a selected metric will be added shown in the table below and will be added to file for downloading.
2. Choose the desired spatial scale for the download. Visible on map will only download data for those reaches currently shown on the map. You can alter the map view using Select view mode on the Map options tab.
3. Click the Download data button to download a csv file of the selected data.
4. Click the Download metadata button to get an html file with information about the downloaded data, including units and the original data source.

The data is provided as a .csv file. If you wish to use this data to make your own maps, then you will need to combine it in a Geographic Information System with the New Zealand digital river network

Change made: Table 1 and instructions on downloading data, and comparing modelled and measured data across environmental categories added to Supplementary file S3.

**References:**

Baisden, W.T., E.D. Keller, R. Van Hale, R.D. Frew, and L.I. Wassenaar. 2016. Precipitation isoscapes for New Zealand: enhanced temporal detail using precipitation-weighted daily climatology. *Isotopes in Environmental and Health Studies* 52: 343-352.

Belachew, D.L., G. Leavesley, O. David, D. Patterson, P. Aggarwal, L. Araguas, S. Terzer, and J. Carlson. 2016. IAEA Isotope-enabled coupled catchment–lake water balance model, IWBMIso: description and validation. *Isotopes in Environmental and Health Studies* 52: 427-442.

Bowen, G.J., C.D. Kennedy, Z. Liu, and J. Stalker. 2011. Water balance model for mean annual hydrogen and oxygen isotope distributions in surface waters of the contiguous United States. *Journal of Geophysical Research: Biogeosciences* 116.

Bowen, G.J., J.B. West, B.H. Vaughn, T.E. Dawson, J.R. Ehleringer, M.L. Fogel, K. Hobson, J. Hoogewerff, C. Kendall, and C.T. Lai. 2009. Isoscapes to address large-scale earth science challenges. *EOS, Transactions American Geophysical Union* 90: 109-110.

Davies-Colley, R.J., D.G. Smith, R.C. Ward, G.G. Bryers, G.B. McBride, J.M. Quinn, and M.R. Scarsbrook. 2011. Twenty Years of New Zealand's National Rivers Water Quality Network: Benefits of Careful Design and Consistent Operation1. *JAWRA Journal of the American Water Resources Association* 47: 750-771.

Fry, B. 2006. *Stable isotope ecology*: Springer.

Halder, J., S. Terzer, L. Wassenaar, L. Araguás-Araguás, and P. Aggarwal. 2015. The Global Network of Isotopes in Rivers (GNIR): integration of water isotopes in watershed observation and riverine research. *Hydrology and Earth System Sciences* 19: 3419-3431.

Hengl, T., G.B.M. Heuvelink, and D.G. Rossiter. 2007. About regression-kriging: From equations to case studies. *Computers & Geosciences* 33: 1301-1315.

Hobson, K.A., and L.I. Wassenaar. 2018. *Tracking animal migration with stable isotopes*: Academic Press.

Julian, J.P., K.M. de Beurs, B. Owsley, R.J. Davies-Colley, and A.G.E. Ausseil. 2017. River water quality changes in New Zealand over 26 years: response to land use intensity. *Hydrol. Earth Syst. Sci.* 21: 1149-1171.

Kerr, T., M. Srinivasan, and J. Rutherford. 2015. Stable water isotopes across a transect of the Southern Alps, New Zealand. *Journal of Hydrometeorology* 16: 702-715.

Keskin, H., and S. Grunwald. 2018. Regression kriging as a workhorse in the digital soil mapper's toolbox. *Geoderma* 326: 22-41.

McDonnell, J.J. 1988. The age, origin and pathway of subsurface stormflow in a steep humid headwater catchment. PhD thesis, University of Canterbury Canterbury, New Zealand.

Purdie, H., N. Bertler, A. Mackintosh, J. Baker, and R. Rhodes. 2010. Isotopic and elemental changes in winter snow accumulation on glaciers in the Southern Alps of New Zealand. *Journal of climate* 23: 4737-4749.

Smith, D.G., and G.B. McBride. 1990. New Zealand's national water quality monitoring network - design and first year's operation. *JAWRA Journal of the American Water Resources Association* 26: 767-775.

Soto, D.X., L.I. Wassenaar, and K.A. Hobson. 2013. Stable hydrogen and oxygen isotopes in aquatic food webs are tracers of diet and provenance. *Functional Ecology* 27: 535-543.

Yang, J., B.D. Dudley, K. Montgomery, and W. Hodgetts. 2020. Characterizing spatial and temporal variation in $^{18}$O and $^{2}$H content of New Zealand river water for better understanding of hydrologic processes. *Hydrological Processes*.